# Joint Semantic Mining for Weakly Supervised RGB-D Salient Object Detection

**Jingjing Li**[1,*], **Wei Ji**[1,*(✉)], **Qi Bi**[2], **Cheng Yan**[3], **Miao Zhang**[4],
**Yongri Piao**[4], **Huchuan Lu**[4,5], **Li Cheng**[1]

[1]University of Alberta, Canada  [2]Wuhan University, China  [3]Tianjin University, China
[4]Dalian University of Technology, China  [5]Pengcheng Lab, Shenzhen, China

## Abstract

Training saliency detection models with weak supervisions, *e.g.,* image-level tags or captions, is appealing as it removes the costly demand of per-pixel annotations. Despite the rapid progress of RGB-D saliency detection in fully-supervised setting, it however remains an unexplored territory when only weak supervision signals are available. This paper is set to tackle the problem of *weakly-supervised RGB-D salient object detection*. The key insight in this effort is the idea of maintaining per-pixel pseudo-labels with iterative refinements by reconciling the multimodal input signals in our joint semantic mining (JSM). Considering the large variations in the raw depth map and the lack of explicit pixel-level supervisions, we propose spatial semantic modeling (SSM) to capture saliency-specific depth cues from the raw depth and produce depth-refined pseudo-labels. Moreover, tags and captions are incorporated via a fill-in-the-blank training in our textual semantic modeling (TSM) to estimate the confidences of competing pseudo-labels. At test time, our model involves only a light-weight sub-network of the training pipeline, *i.e.*, it requires only an RGB image as input, thus allowing efficient inference. Extensive evaluations demonstrate the effectiveness of our approach under the weakly-supervised setting. Importantly, our method could also be adapted to work in both fully-supervised and unsupervised paradigms. In each of these scenarios, superior performance has been attained by our approach with comparing to the state-of-the-art dedicated methods. As a by-product, a *CapS* dataset is constructed by augmenting existing benchmark training set with additional image tags and captions. *Code and dataset are available at* `https://github.com/jiwei0921/JSM`.

## 1 Introduction

As a fundamental computer vision task, salient object detection (SOD) aims at locating and segmenting visually distinctive objects in a scene. It plays an important role in a variety of downstream applications including image retrieval [31, 65], medical analysis [48, 28, 25], multimodal fusion [79, 80] and video analysis [88, 81, 90]. Recent progress in supervised RGB-D SOD [5, 52, 10, 26, 89, 33] has demonstrated significant benefits of engaging depth information in saliency detection from complex scenes. The success of these fully-supervised methods, however, relies heavily on the large-scale, precise, pixel-level annotations, which are often laborious and time-consuming to acquire. On the other hand, an image usually comes with additional information in its meta-data such as tags and captions from users to describe the scene context and content, which may serve as cheap weak-supervision signals. These weak supervision signals are nonetheless noisy and have mixed qualities. Similar weak-supervision signals have been explored in RGB-image based SOD [63, 70, 38], where the noisy nature of these side information is unfortunately overlooked. To further complicate the matter, the lack of explicit pixel-level supervision brings new challenge to the RGB-D SOD task: the depth values from raw depth maps are often noisy and sometimes inconsistent. For example, in Fig. 1,

---

*means equal contributions. Wei Ji ✉ (wji3@ualberta.ca) is the corresponding author.

35th Conference on Neural Information Processing Systems (NeurIPS 2021)

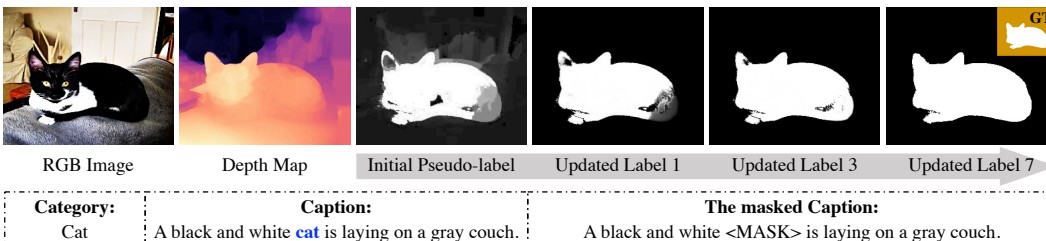

| Category: | Caption: | The masked Caption: |
|---|---|---|
| Cat | A black and white **cat** is laying on a gray couch. | A black and white \<MASK\> is laying on a gray couch. |

Figure 1: Illustration of weakly-supervised RGB-D salient object detection. RGB and depth images, as well as weak supervision signals such as image-level tags and captions are exploited. Initial pseudo-label is generated by the handcrafted methods, which is then iteratively updated by our joint semantic mining pipeline. GT is ground-truth label for reference.

similar depth values are shared by the cat and the underneath couch, making it difficult to discern the salient object from backgrounds. Without the explicit pixel-level supervision, existing cross-modal fusion strategies adopted by fully-supervised RGB-D methods [36, 55, 37, 27] would simply fail.

These observations motivate us to consider the new problem of *weakly-supervised RGB-D salient object detection*, which takes as input the RGB and depth images, as well as weak supervision signals such as image-level tags and captions, as illustrated in Fig. 1. By removing the demand for laborious per-pixel annotations, it also brings new challenges: 1) how to address the noisy nature of the weak supervision signals; 2) how to tackle the depth noise and inconsistency to facilitate proper separation of foreground and background regions.

This leads us to propose the use of pseudo-labels with iterative refinements in training: the pseudo-label provides internal pixel-level supervision signals, which is progressively updated by reconciling the multimodal input signals and the current information flow of the neural net, based on the previous pseudo-label. As illustrated in Fig. 2, this is realized by an interaction between two core modules, namely spatial semantic modeling (SSM) and joint semantic mining (JSM): SSM is designed to capture the saliency-specific depth semantics, to eliminate the background noises in the coarse saliency prediction, and to generate a depth-refined pseudo-label. This simple yet effective module is very generic, which could be easily plugged-in different setups, including unsupervised & fully-supervised scenarios; meanwhile, the JSM module is proposed to leverage depth semantics and weak supervision signals for attaining more reliable pseudo-labels. Specifically, a partial textual input, *i.e.,* image-level tag and caption with its salient word being masked, is fed into a dedicated textual semantic modeling or TSM to estimate the confidence scores of competing pseudo-labels, and to fill-in-the-blank. Intuitively, a semantically consistent pseudo-label should provide better context cues to reconstruct the salient word; while a closer guess of the masked word would indicate a better pseudo-label. The alternation between SSM and JSM modules is thus expected to give rise to more trustworthy pseudo-labels. At test time, it is then sufficient to take an input RGB image and activate a light-weight network to deliver its final prediction. That is, test time input involves only an RGB image, without the need of any depth map or image-level tags and captions. This drastically simplifies the input requirement and reduces the computation burden at deployment stage. Moreover, given the lack of training dataset for the weakly-supervised setting, we adapt existing RGB-D training dataset to annotate additional image-level tags and captions, which is referred to as the *CapS* dataset.

The main contributions of this paper are as follows. (1) A new problem of weakly-supervised RGB-D salient object detection is considered. In this regard, a *CapS* dataset is curated by augmenting the existing RGB-D SOD training dataset with image-level tagging and captioning annotations. (2) The key ingredient of our approach involves the production of pseudo-labels with iterative refinements, realized by iterative updates between two internal modules, SSM and JSM. Empirical experiments demonstrate the effectiveness of our approach in weakly-supervised setting. Moreover, (3) after proper adaptation of our approach in unsupervised and fully-supervised scenarios, superior performance is also observed when comparing to the respective state-of-the-art methods. (4) Our test time inference amounts to executing a light-weight saliency network: as illustrated in dotted line at Fig. 2, only an RGB image is used as its input, thus allows efficient and effective inference.

## 2   Related Work

RGB-D salient object detection [87, 93, 94] has been an active field of research in the past few years, where the incorporation of depth cues has been demonstrated [13, 34, 82, 75, 15, 35, 60, 43, 85]

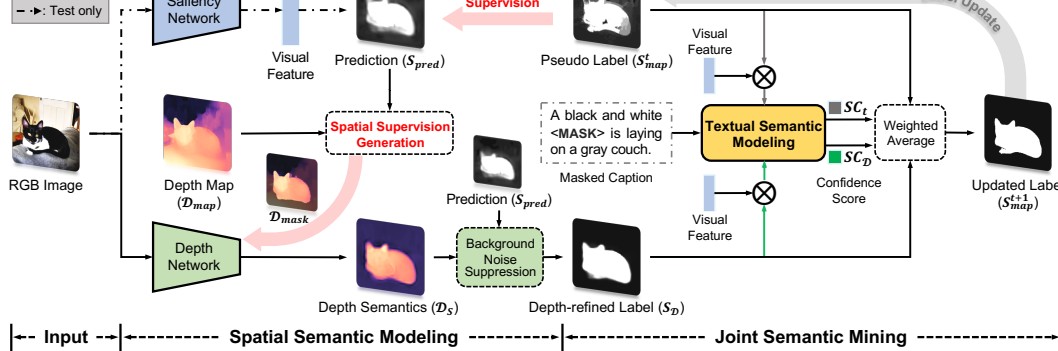

Figure 2: An overview of our approach. Its training pipeline consists of a saliency prediction network, a SSM (Sec. 3.2) to generate depth-refined pseudo-label, a TSM (Sec. 3.3) to estimate the confidences of different pseudo-labels, and a JSM (Sec. 3.4) to refine & update pseudo-label. Our testing process only involves activating a saliency network delineated in dotted lines. More details of SSM and TSM modules are illustrated in Fig. 3. The masked salient word is 'cat'.

to improve performance especially in complex scenes. Existing RGB-D methods aim to design effective feature fusion strategies for learning representative cross-modal features. Typically, Chen *et al.* [8] employ two-stream CNNs-based models and perform fusion by adding or concatenating paired features at shallow or deep layers. Fu *et al.* [19] utilize a Siamese network to jointly learn RGB and depth inputs for mining useful complementary features. To promote multi-modal interactions, Li *et al.* [37] design a cross-modal weighting strategy to encourage comprehensive interactions between RGB and depth information. However, those methods unfortunately rely on costly pixel-level annotations, which are tedious and time-consuming to acquire. This motivates us to consider a weakly-supervised approach. In what follows, our focus will be mainly toward related weakly-supervised methods developed for saliency detection from RGB images, where the differences of our approach from existing methods would be clarified.

Instead of using costly pixel-level annotations, some recent efforts instead explore cheap alternatives such as image-level tags (categories) [38, 63, 71, 46, 2], image captions [70], scribble labels [69, 76], and noisy pseudo-labels from handcrafted methods [45, 72, 73, 77, 68, 47, 67, 49, 66]. The pioneering work [63] leverage image-level tags or categories that could be augmented onto existing large-scale dataset at low-cost. The same scenario is also considered by Li *et al.* [38], where a composite pipeline combining graphical model with CNNs is designed. The trained network however tends to highlight the most discriminative region instead of the intended salient object out of the scene due to the sparse image-level supervisions. Image captions are examined by Zeng *et al.* [70] as supervision input; in their work image classification network and caption generation network are jointly trained to obtain pseudo-labels, which achieves descent performance. Scribble is another type of weak supervision signal, where a tiny fraction of image pixels are labeled by users as being foreground or background. Due to the annotation sparsity, object structure and details cannot be easily inferred. Zhang *et al.* [76] introduce a gated structure-aware loss as well as an auxiliary edge detection network to uncover the complete object. Meanwhile, Yu *et al.* [69] explore self-consistency among multi-scale outputs and design a local coherence loss to propagate the labels to unlabeled regions based on image features, thus enabling the detection of objects with smooth textures. However, existing weakly supervised saliency methods are solely based on RGB image. Unlike the prevalence of fully-supervised RGB-D SOD, it has never been considered the incorporation of depth cues in existing literature.

This leads us to address this problem in the presence of image tags and captions as weak supervision signals. Different from existing methods [38, 24, 63, 70] that train image classification networks or caption generation networks to delineate potential salient regions, masked salient word in captions are to be reconstructed in our work by leveraging visual saliency features. This is then used to estimate the confidence scores of pseudo-labels.

## 3 Methodology

### 3.1 The Overall Architecture

An overview of our approach is illustrated in Fig. 2. It consists of a saliency network responsible for saliency prediction, a spatial (depth) semantic modeling (SSM) to generate depth-refined pseudo-label,

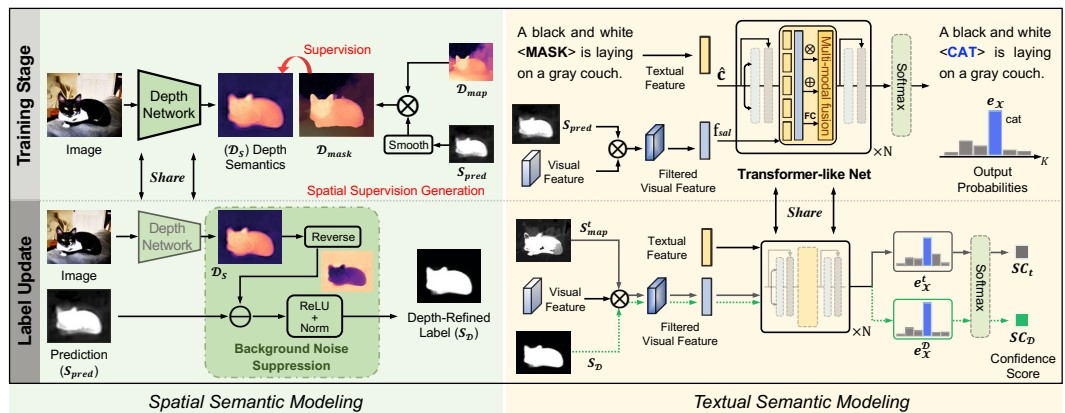

Figure 3: The detailed architecture of the proposed SSM and TSM. The upper shows their training processes, and the bottom illustrates the way of using them to perform label update.

a textual (caption) semantic modeling (TSM) to estimate the confidences of different pseudo-labels, and a joint semantic mining (JSM) strategy to refine & update pseudo-label. Overall our pipeline aims to gradually improve the quality of noisy pseudo-labels by jointly mining the useful spatial semantics from the depth map and textual semantics from the tags and captions, to produce more trustworthy supervision signals, which in turns results in better training of the saliency network.

Specifically, the popular encoder-decoder architecture [64] in SOD is adopted in both saliency network and depth network. Initial supervision signal for the saliency network is provided by traditional handcrafted methods. The predicted saliency map, together with the raw depth map, are processed to generate the saliency-guided spatial supervision signal for the depth network. Then the predicted saliency-oriented depth semantics is utilized to eliminate background noises (non-salient regions) in coarse prediction, and to generate a depth-refined pseudo-label. This is followed by our JSM strategy, which takes in the image-level tags and captions through TSM to estimate the confidence scores of pseudo-labels; updated pseudo-label is then formed based on the confidence-weighted depth-refined pseudo-label and current pseudo-label, which provides more trustworthy supervision signal for the saliency network. Note our test time inference involves only the black dashed portion in Fig. 2, which takes as input only the RGB image, thus enables efficient saliency prediction.

## 3.2 Saliency-oriented Spatial (Depth) Semantic Modeling

Initial pseudo-labels are generated by traditional saliency models, which often contain excessive noise. As illustrated in Fig. 3, our spatial semantic modeling (SSM) is to produce a more reliable depth-refined pseudo-label, achieved by explicitly capturing saliency-specific depth semantics from the depth map to eliminate possible background noise in the coarse saliency prediction.

Concretely, during training, we first generate a saliency-guided depth mask $\mathcal{D}_{mask}$ by multiplying the rough saliency prediction $\mathcal{S}_{pred}$ with the raw depth map $\mathcal{D}_{map}$ in a spatial attention manner. Here, a Gaussian smooth operation is applied to smooth the predicted saliency area, to effectively perceive and capture more saliency areas from depth. The procedure is formulated as:

$$\mathcal{D}_{mask} = \Omega_{max}(\mathcal{F}_{gauss}(\mathcal{S}_{pred}, k), \mathcal{S}_{pred}) \otimes \mathcal{D}_{map}, \tag{1}$$

where $\mathcal{F}_{gauss}(\cdot, k)$ indicates a convolution operation with Gaussian kernel $k$ and zero bias; $\Omega_{max}(\cdot, \cdot)$ is a maximum function to preserve the higher values between the smoothed and the original maps. $\otimes$ is element-wise multiplication. In this paper, the size and standard deviation of kernel $k$ are learnable through the model training procedure and are initialized with values 32 and 4, respectively.

After obtaining $\mathcal{D}_{mask}$, a depth network is trained to learn the saliency-specific depth semantics $\mathcal{D}_{\mathcal{S}}$, using the mean square error (MSE) loss function. The internal inspection evidences in Fig. 4 suggest that $\mathcal{D}_{\mathcal{S}}$ (depth semantics, $6^{th}$ column) is able to capture discriminative saliency cues from the raw depth map under the supervision of $\mathcal{D}_{mask}$ ($5^{th}$ column).

In addition, the learned $\mathcal{D}_{\mathcal{S}}$ can be further processed to generate the depth-refined pseudo-labels. This procedure is only employed when performing pseudo-label update. Specifically, we feed $\mathcal{D}_{\mathcal{S}}$ into an Background Noise Suppression block to help eliminate the background noises (non-salient

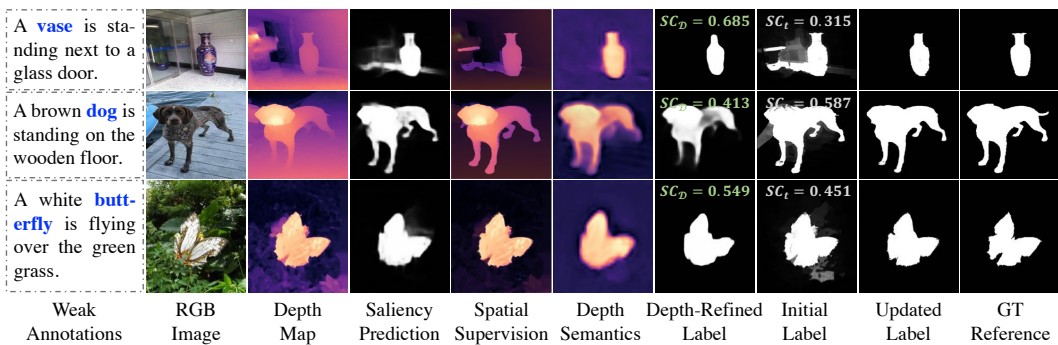

| Weak Annotations | RGB Image | Depth Map | Saliency Prediction | Spatial Supervision | Depth Semantics | Depth-Refined Label | Initial Label | Updated Label | GT Reference |

Figure 4: Step-by-step inspections of the internal processes of our approach. The GT is for reference.

regions) in the coarse prediction $\mathcal{S}_{pred}$. In this block, a reverse operation is first employed on $\mathcal{D}_{\mathcal{S}}$ to highlight background regions by $1 - \mathcal{D}_{\mathcal{S}}$. This is followed by the pixel-wise subtraction to suppress the non-salient negative responses in $\mathcal{S}_{pred}$. Finally, a ReLU function and a normalization procedure are adopted to suppress negative numbers and normalize the result to the range of [0, 1]. This procedure of obtaining the depth-refined pseudo-label $\mathcal{S}_{\mathcal{D}}$ could be expressed by

$$\mathcal{S}_{\mathcal{D}}^{i,j} = \frac{\mathcal{S}_d^{i,j} - min(\mathcal{S}_d)}{max(\mathcal{S}_d) - min(\mathcal{S}_d) + \varepsilon}, \ \mathcal{S}_d = \mathrm{ReLU}(\mathcal{S}_{pred} - \lambda_d(1 - \mathcal{D}_{\mathcal{S}})), \ i \in [1, H], j \in [1, W]. \ (2)$$

Here $\lambda_d \in [0, 1]$ is a constant to control the degree of the subtracted background noises and avoid negatively suppressing salient regions that have relatively low saliency scores in $\mathcal{S}_{\mathcal{D}}$. Throughout experiments, $\lambda_d$ is empirically set to 0.5, and $\varepsilon$ to 1e-5. $H$ and $W$ are the height and width of the input image, respectively.

The pseudo-label refinement dynamics are visually inspected in Fig. 4, while Fig. 8 presents the corresponding quantitative error analysis over iterations. Empirical evidence suggests as the training proceeds, quality of the pseudo-label is significantly improved. Moreover, at this stage, we can directly utilize the depth-refined label $\mathcal{S}_{\mathcal{D}}$ to update pseudo-labels (as in Fig. 6 (b)). This can be referred as unsupervised RGB-D SOD since no weak labels are used.

### 3.3 Saliency-oriented Textual (Caption) Semantic Modeling

Previously, the mainstream use of weak labels is to train a classification network or a caption generation network, where the by-product attention maps or Class Activation Maps [92] are leveraged to determine the potential salient regions [70, 63]. It is very different in our textual semantic modeling (TSM), where the main focus is to leverage side information (*i.e.*, image-level tags and captions) to facilitate the production of reliable training signals. Inspired by the recent success of masked language models [14], captions with missing keywords are used as input, with the expectation of the complete text being reconstructed as output. In the proposed TSM, innovatively taking as input partial text with salient word being masked, as well as the saliency-filtered visual features, our TSM is to output the reconstructed text in a fill-in-the-blank manner and to estimate the confidence scores of competing pseudo-labels. The intuition is, a semantically matching pseudo-label could provide better context cues to reconstruct the masked salient word; meanwhile, a closer guess of the masked word would indicate a better pseudo-label.

Formally, for each training data, the weak labels contain caption description $\mathbf{c} = \{\mathbf{c}_i\}_{i=1}^{n_c}$, image-level category $k \in \{1, .., K\}$, and the position $\mathcal{X}$ of the salient word (object) in the caption, where $n_c$ is the word number of the caption. Let $\mathbf{c}_i \in \mathbb{R}^{d \times 1}$ be the word embedding of the $i$-th word in the caption, $K$ the total number of salient categories, and $\mathcal{X}$ an integral number. As shown in Fig. 3, during training, the input is a masked version of caption $\hat{\mathbf{c}} \in \mathbb{R}^{d \times n_c}$ where the salient word $\mathbf{c}_{\mathcal{X}}$ in caption $\mathbf{c}$ is masked with a special symbol. In order to reconstruct the masked salient word, we filter the visual feature from the saliency network by multiplying it with the learned saliency cues $\mathcal{S}_{pred}$. We then obtain the saliency-filtered visual feature, and transform it to a feature vector $\mathbf{f}_{sal} \in \mathbb{R}^{d \times 1}$ for subsequent cross-modal fusion using a pooling operation and a fully-connected (FC) layer.

The center component of the TSM module is a transformer-like network. Based on original Transformer [62], we add a multi-modal fusion sub-layer into network. Three parallel operations are used to promote sufficient cross-modal feature interactions: element-wise multiplication, element-wise

addition, and concatenation followed by FC. The three outputs are concatenated and followed by a FC to change the feature dimension. Note the cross-modal fusion operation is performed word-wise. Through the textual network, we can obtain the energy vector $\mathbf{e}_\mathcal{X} \in \mathbb{R}^{K \times 1}$ of the masked salient word, which is computed over all categories by a fully-connected layer and softmax function. For each training sample, the training objective loss for the textual network is $-\log(\mathbf{e}_\mathcal{X}[k])$, where $\mathbf{e}_\mathcal{X}[k]$ is the output probability of salient category $k$. *Detailed structure for our TSM can be accessed in the supplementary material.* Once trained, the TSM module is then used to estimate the confidence scores of pseudo-labels as described in Sec. 3.4.

### 3.4 Joint Semantic Mining for Label Updating

The label updating operation using joint semantic mining is iteratively conducted every $\tau$ epochs (as a training round) during training, *i.e.*, the granularity of label updating. The choices are explored in the ablations of Sec. 4.4, where $\tau = 5$ is shown to work best empirically. In terms of training, the saliency network, the SSM module, and the TSM module are independently trained.

At the end of each training round, our JSM strategy performs the label updating operation to generate an up-to-date pseudo-label for each training image. Now define the pseudo-label for the saliency network in current iteration as $\mathcal{S}_{map}^t$. As shown in the label update phase of Fig. 3, the SSM module is engaged to generate the depth-refined pseudo-label $\mathcal{S}_\mathcal{D}$; It is passed to the TSM module, where the $\mathcal{S}_{map}^t$ and $\mathcal{S}_\mathcal{D}$ are taken as saliency attention maps to filter visual features, respectively. The TSM module infers the energy vectors $\mathbf{e}_\mathcal{X}^t$ and $\mathbf{e}_\mathcal{X}^\mathcal{D}$ for $\mathcal{S}_{map}^t$ and $\mathcal{S}_\mathcal{D}$, respectively. Thus their confidence scores $\mathcal{SC}_t$ and $\mathcal{SC}_\mathcal{D}$ can be calculated by (taking $\mathcal{SC}_t$ as an example):

$$\mathcal{SC}_t = \frac{\exp(\mathbf{e}_\mathcal{X}^t[k])}{\exp(\mathbf{e}_\mathcal{X}^t[k]) + \exp(\mathbf{e}_\mathcal{X}^\mathcal{D}[k])}. \tag{3}$$

The updated label is the weighted average of $S_{map}^t$ and $\mathcal{S}_\mathcal{D}$: $\mathcal{S}_{map}^{t+1} = \mathcal{SC}_t \times \mathcal{S}_{map}^t + \mathcal{SC}_\mathcal{D} \times \mathcal{S}_\mathcal{D}$. A fully-connected conditional random field [23] is applied to generate the final updated label which could provide more trustworthy supervision signal to train the saliency network.

### 3.5 The CapS Dataset

To train the weakly-supervised RGB-D SOD network, we relabel two widely-used RGB-D saliency datasets, NJUD [29] and NLPR [53], which contain a total of 2,185 training images. We provide various image-level weak annotations: categories, captions, and position of the salient word in each of the captions. The annotation process is conducted semi-automatically: the NeuralTalk2 [30] toolkit is utilized to automatically generate image captions, which are then manually checked, with unreasonable cases corrected. On average, there are 8.9 words in each caption. This is followed by categorically tagging each of the images, each corresponds to a salient category; this is different from the traditional image classification dataset ImageNet [32]. We summarize the 100 categories in the *CapS*. The positions of the salient words are first automatically identified through localizing the image categories in the captions which is subsequently followed by manual identification. *Detailed statistics and examples of our in-house CapS dataset are relegated to the supplementary material.*

## 4 Experiments

### 4.1 Datasets and Evaluation Metrics

Empirical evaluations are conducted over four large-scale RGB-D SOD benchmark datasets, including NJUD [29] with 1,985 RGB-D paired images, NLPR [53] with 1,000 samples, STERE [50] with 1,000 stereoscopic images, and DUTLF-Depth [55] with 1,200 RGB-D data. In train *vs.* test splits of the datasets, the popular setup of [18, 19, 74] is followed for a fair comparison. Training set consists of 1,485 samples from NJUD and 700 samples from NLPR. The remaining images in these datasets and other public test sets are reserved for testing purposes throughout the experiments. Four widely-used metrics are adopted for quantitative evaluation: they are E-measure ($E_\xi$) [16], F-measure ($F_\beta$) [1], weighted F-measure ($F_\beta^w$) [44], and mean absolute error (MAE or $\mathcal{M}$) [3].

### 4.2 Implementation Details and Setups

The code is implemented in Pytorch toolbox on a PC with a single Tesla P40 GPU. We use decoder part [64] with ResNet-50 [22] pre-trained on ImageNet as backbone, for both saliency network and depth network. For each word in the caption, we extract word embedding with dimension $d = 300$ using the pretrained Glove [54] word2vec network. The maximum caption length is set to 20. For

Table 1: Quantitative comparison with weakly-supervised and unsupervised saliency models. Note RGB-based methods are specifically marked by ‡. $\mathcal{D}_S$ and $\mathcal{T}_S$ represent the spatial semantic modeling and textual semantic modeling, respectively. 'Un' means unsupervised learning. 'Cls' is SOD with class label. 'Cap' represents using weak supervision signals, with both class label and image caption.

| * | Sup. | DUTLF-Depth [55] | | | | STERE [50] | | | | NJUD [29] | | | | NLPR [53] | | | |
|---|---|---|---|---|---|---|---|---|---|---|---|---|---|---|---|---|---|
| | | $E_\xi$ | $F_\beta^w$ | $F_\beta$ | $\mathcal{M}$ | $E_\xi$ | $F_\beta^w$ | $F_\beta$ | $\mathcal{M}$ | $E_\xi$ | $F_\beta^w$ | $F_\beta$ | $\mathcal{M}$ | $E_\xi$ | $F_\beta^w$ | $F_\beta$ | $\mathcal{M}$ |
| RBD‡ [98] | Un | .733 | .447 | .619 | .222 | .730 | .443 | .610 | .223 | .684 | .387 | .556 | .256 | .765 | .388 | .590 | .211 |
| MST‡ [61] | Un | .678 | .254 | .401 | .279 | .681 | .312 | .447 | .269 | .670 | .291 | .436 | .281 | .762 | .257 | .491 | .199 |
| BSCA‡ [57] | Un | .808 | .479 | .682 | .181 | .803 | .497 | .676 | .179 | .756 | .446 | .623 | .216 | .745 | .376 | .554 | .178 |
| DSR‡ [39] | Un | .797 | .478 | .640 | .164 | .785 | .486 | .645 | .165 | .739 | .436 | .594 | .196 | .757 | .451 | .545 | .120 |
| ACSD [29] | Un | .250 | .210 | .188 | .668 | .793 | .425 | .661 | .200 | .790 | .448 | .696 | .198 | .751 | .327 | .547 | .171 |
| DES [11] | Un | .733 | .386 | .668 | .280 | .673 | .383 | .592 | .297 | .421 | .241 | .165 | .448 | .735 | .259 | .583 | .301 |
| LHM [53] | Un | .767 | .350 | .659 | .174 | .772 | .360 | .703 | .171 | .722 | .311 | .625 | .201 | .772 | .320 | .520 | .119 |
| GP [59] | Un | - | - | - | - | .785 | .371 | .710 | .182 | .730 | .323 | .666 | .204 | .813 | .347 | .670 | .144 |
| CDB [40] | Un | - | - | - | - | .808 | .436 | .713 | .166 | .752 | .408 | .650 | .200 | .810 | .388 | .618 | .108 |
| SE [20] | Un | .730 | .339 | .474 | .196 | .825 | .546 | .747 | .143 | .780 | .518 | .735 | .164 | .853 | .578 | .701 | .085 |
| DCMC [12] | Un | .712 | .290 | .406 | .243 | .832 | .529 | .743 | .148 | .796 | .506 | .715 | .167 | .684 | .265 | .328 | .196 |
| MB [96] | Un | .691 | .464 | .577 | .156 | .693 | .455 | .572 | .178 | .643 | .369 | .492 | .202 | .814 | .574 | .637 | .089 |
| CDCP [97] | Un | .794 | .530 | .633 | .159 | .797 | .596 | .666 | .149 | .751 | .522 | .618 | .181 | .785 | .512 | .591 | .114 |
| **Ours** ($\mathcal{D}_S$) | Un | **.845** | **.629** | **.741** | **.116** | **.838** | **.654** | **.738** | **.111** | **.781** | **.576** | **.689** | **.147** | **.857** | **.638** | **.698** | **.073** |
| WSS‡ [63] | Cls | .843 | .634 | .743 | .125 | .831 | .664 | .732 | .115 | .760 | .566 | .693 | .149 | .856 | .651 | .712 | .077 |
| MSW‡ [70] | Cap | .863 | .678 | .789 | .105 | .836 | .675 | .760 | .103 | .773 | .580 | .704 | .141 | .869 | .659 | .736 | .071 |
| **Ours** ($\mathcal{D}_S\&\mathcal{T}_S$) | Cap | **.870** | **.698** | **.797** | **.093** | **.852** | **.688** | **.778** | **.095** | **.788** | **.599** | **.717** | **.133** | **.888** | **.692** | **.770** | **.060** |

Image  Depth  GT  Ours  MSW  CDCP  LHM  DCMC  DSR  BSCA

Figure 5: Visual comparison of weakly-supervised and unsupervised saliency models.

the transformer-like net in TSM, we set the hidden state to 256, the number of layers to 3, and the number of head to 4. The model is optimized by Adam with batch size of 10, and the learning rate is set to $1 \times 10^{-4}$. During training, we use the standard BCE loss to train the saliency network. Each image is uniformly resized to $352 \times 352$ and is performed by randomly rotating and cropping to avoid potential overfitting. Our network is trained in an end-to-end manner and converges around 50 epochs. Initial pseudo labels are generated by the handcrafted method [97], which is freely available for annotations.

## 4.3 Model Performance

We quantitatively evaluate the performance of our approach in Table 1, with visual results shown in Fig. 5. Since our approach is the first work for weakly-supervised RGB-D saliency detection, we show the results of two state-of-the-art RGB-based weakly-supervised methods (WSS [63] and MSW [70]) for reference. To make a fair comparison, we fine-tune them on the same training set using their published code with default setups. These results show the effectiveness of our proposed method. Furthermore, the SSM and TSM in our joint semantic mining framework do not introduce any additional inference cost since they only participate in the training procedure to provide more reliable supervisory signals for saliency network. This design leads to a light-weight network that works both efficiently and effectively. As shown in Table 2, our method runs fastest among RGB-D methods, and also on par with the most efficient RGB-based methods.

Table 2: Inference time of different unsupervised and weakly-supervised saliency models. The RGB-based methods are specifically marked by ‡.

| * | LHM [53] | DES [11] | GP [59] | CDCP [97] | DCMC [12] | SE [20] | ACSD [29] | CDB [40] | BSCA‡ [57] | RBD‡ [98] | MST‡ [61] | DSR‡ [39] | MSW‡ [70] | **Ours** |
|---|---|---|---|---|---|---|---|---|---|---|---|---|---|---|
| Inference Time (s) | 2.13 | 7.79 | 12.98 | 5.7 | 1.2 | 1.57 | 0.718 | 0.6 | 2.665 | 0.1893 | 0.0302 | 0.3758 | 0.0267 | 0.0286 |

## 4.4 Empirical Analysis of Our Pipeline

Here we focus on evaluating the contribution of each component in our pipeline, and examining the internal performance of the pseudo-labels at different stages in training.

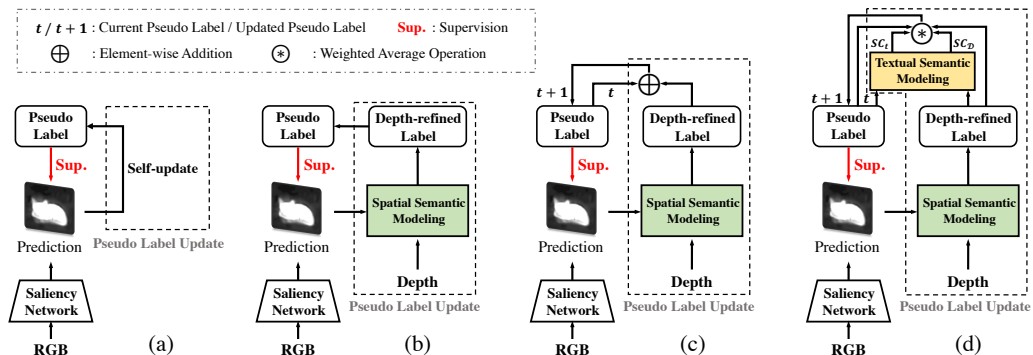

**Figure 6:** Diagrams of various label updating strategies used in our ablation study: (a) self-updating strategy, (b) SSM updating strategy, (c) historical moving average with equal weights, and finally (d) our JSM strategy.

**Table 3:** Ablation study of our pipeline. ↑ (↓) denote performance gains (relative to backbone).

| Model Setups | DUTLF-Depth [55] | | | STERE [50] | | | NJUD [29] | | | NLPR [53] | | |
|---|---|---|---|---|---|---|---|---|---|---|---|---|
| | $F_\beta^w$ | $F_\beta$ | $\mathcal{M}$ | $F_\beta^w$ | $F_\beta$ | $\mathcal{M}$ | $F_\beta^w$ | $F_\beta$ | $\mathcal{M}$ | $F_\beta^w$ | $F_\beta$ | $\mathcal{M}$ |
| Backbone trained on Pseudo Labels (*i.e.*, '$\mathcal{B}$') | .512 | .644 | .167 | .555 | .666 | .158 | .510 | .627 | .186 | .479 | .570 | .126 |
| '$\mathcal{B}$' trained on Pseudo Labels with CRF | .568 | .670 | .140 | .601 | .684 | .135 | .546 | .642 | .165 | .585 | .624 | .094 |
| '$\mathcal{B}$' with Self-updating Strategy | .616 | .697 | .130 | .643 | .708 | .123 | .571 | .673 | .154 | .607 | .651 | .087 |
| '$\mathcal{B}$' with Spatial Semantic Modeling | .629 | .741 | .116 | .654 | .738 | .111 | .576 | .689 | .147 | .638 | .698 | .073 |
| | ↑23% | ↑15% | ↓31% | ↑18% | ↑11% | ↓30% | ↑13% | ↑10% | ↓21% | ↑33% | ↑22% | ↓42% |
| '$\mathcal{B}$' with SSM and Historical Moving Average | .644 | .750 | .113 | .674 | .753 | .105 | .588 | .698 | .141 | .664 | .722 | .068 |
| '$\mathcal{B}$' with Joint Semantic Mining (*i.e.*, **Ours**) | **.698** | **.797** | **.093** | **.688** | **.778** | **.095** | **.599** | **.717** | **.133** | **.692** | **.770** | **.060** |
| | ↑36% | ↑24% | ↓44% | ↑24% | ↑17% | ↓40% | ↑17% | ↑14% | ↓28% | ↑44% | ↑35% | ↓52% |

**Ablation analysis.** We present in Table 3 the ablation results of our pipeline on four benchmarks. To start with, we consider the backbone as the saliency network trained with initial pseudo labels. As our proposed SSM and TSM are gradually incorporated into the backbone to generate the depth-refined pseudo-labels and estimate their confidence scores for label updating, noticeable performance gains are consistently achieved in all datasets. The SSM significantly reduces the MAE metric and increases the F-measure score by 31% and 14.5% on average in four datasets, respectively. The TSM further boosts the saliency detection performance to a higher level where a significant amount of performance gains with 41% and 22.5% are finally achieved on MAE and F-measure metrics.

To further demonstrate the effectiveness of our SSM in exploiting the depth semantics to refine pseudo-labels, we retrain the saliency network using the self-updating strategy as in Fig. 6 (a). In this strategy, the saliency prediction with CRF are directly utilized to update pseudo-labels, at the end of each training round. As shown in Table 3 ($3^{rd}$ row *vs.* $4^{th}$ row), when excluding saliency-oriented depth semantics captured by the SSM, the performance of model degrades greatly. This indicates our SSM can effectively suppress background noise and providing reliable training labels. Furthermore, we replace TSM with a heuristic historical moving average strategy where the pseudo-labels are assigned with equal weights as in Fig. 6 (c). Table 3 shows it achieves better performance than the backbone using SSM due to the consideration of historical information, but it is consistently inferior compared to our pipeline with the TSM module. These results suggest that our TSM can better estimate the confidence or quality of pseudo-labels and generate the trustworthy supervision signals. Meanwhile, we also provide the internal inspections of our approach in Fig. 4, in terms of the generation of pseudo-labels and their corresponding confidence scores, for better understanding.

In addition, we further discuss the effect of different update intervals in our pipeline when performing label updating. As listed in Table. 4, the larger or smaller update interval leads to inferior performance due to the insufficient or excessive learning of models.

**Table 4:** Parameter analysis of the update interval $\tau$ (epoch) in the JSM.

| Interval ($\tau$) | STERE [50] | | NJUD [29] | | NLPR [53] | |
|---|---|---|---|---|---|---|
| | $F_\beta$ | $\mathcal{M}$ | $F_\beta$ | $\mathcal{M}$ | $F_\beta$ | $\mathcal{M}$ |
| $\tau = 1$ | .769 | .098 | .703 | .142 | .766 | .063 |
| $\tau = 5$ | .778 | .095 | .717 | .133 | .770 | .060 |
| $\tau = 10$ | .758 | .101 | .695 | .138 | .763 | .065 |

**Analysis of pseudo-labels.** We analyze the evolutions of pseudo-labels in our training process. Here the quality of pseudo-labels is measured using the ground-truth labels of the training set (for evaluation only). Visual evidences in Fig. 7 show that the quality of pseudo-labels is gradually improved as our JSM is performed. We can see that the initial pseudo-labels unfortunately tend to miss important salient parts as well as fine-grained details. By adopting our proposed joint semantic mining for label updating, the missing parts could be gradually retrieved, with the object silhouette

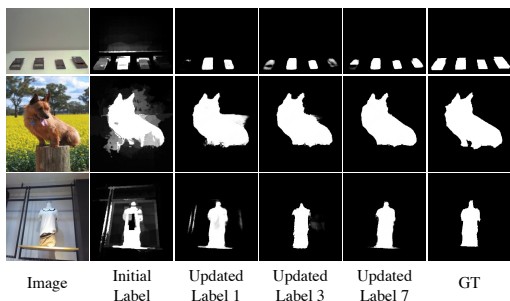

| Image | Initial Label | Updated Label 1 | Updated Label 3 | Updated Label 7 | GT |

Figure 7: Visualization of updated labels.

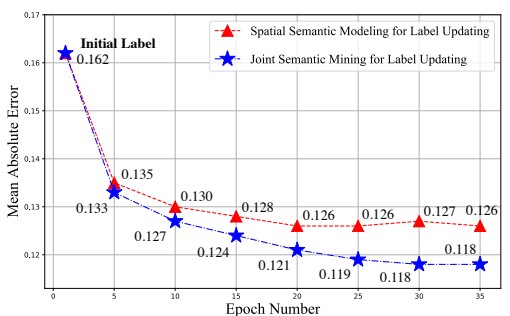

Figure 8: Error reduction plot of the updated pseudo-labels over iterations.

Table 5: Ablation analysis of the supervised variant of our approach, where the human annotations (*i.e.,* ground truths) are used to train SOD models.

| Model Setups (fully) | STERE [50] | | | NJUD [29] | | | NLPR [53] | | |
|---|---|---|---|---|---|---|---|---|---|
| | $F_\beta^w$ | $F_\beta$ | $\mathcal{M}$ | $F_\beta^w$ | $F_\beta$ | $\mathcal{M}$ | $F_\beta^w$ | $F_\beta$ | $\mathcal{M}$ |
| Backbone (*i.e.,* '$\mathcal{B}$') | .858 | .869 | .045 | .864 | .871 | .046 | .865 | .863 | .028 |
| '$\mathcal{B}$' + SSM (Ours) | .876 | .896 | .039 | .885 | .906 | .038 | .892 | .905 | .022 |

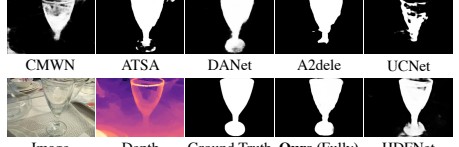

| CMWN | ATSA | DANet | A2dele | UCNet |
| Image | Depth | Ground Truth | **Ours** (Fully) | HDFNet |

Figure 9: Visual comparison of fully supervised RGB-D SOD methods.

also being refined. The final pseudo-label is closest to the true label, which could provide more reliable guiding signal for training the saliency network. Moreover, we present in Fig. 8 the error reduction curves of the updated pseudo-labels with our full pipeline (blue line) and SSM only (red line), respectively. It can be seen that only SSM is able to improve the quality of pseudo-labels by exploiting the useful depth semantics to refine pseudo-labels. Our JSM further boosts the performance by leveraging the textual semantics to integrate reliable pseudo-labels.

## 4.5 Generalization Analysis

**Adapting to unsupervised setting.** Our approach can be adapted to unsupervised setting by using the architecture illustrated in Fig. 6 (b), *i.e.*, SSM, where only the depth semantics are mined to refine pseudo-labels without weak labels. The quantitative results in Table 1 show the effectiveness of our SSM in unsupervised setting.

**Adapting to fully-supervised setting.** Apart from the adaption to unsupervised setting, a variant of our approach can also be applied to fully-supervised RGB-D SOD scenario. This is achieved by modifying the generation of saliency-guided depth mask as $\mathcal{D}_{mask} = \mathcal{S}_{GT} \otimes \mathcal{D}_{map}$ in Eq. 1, with $\mathcal{S}_{GT}$ being the ground-truth annotations. The saliency network and depth network are trained by $\mathcal{S}_{GT}$ and $\mathcal{D}_{mask}$, respectively. Then the background noise suppression block in SSM module is applied to obtain the final saliency. As ablated in Table 5, our fully-supervised variant achieves consistent performance improvement compared to the backbone trained on $\mathcal{S}_{GT}$. In addition, our results compare favorably with those of 21 fully-supervised RGB-D saliency models, as quantitatively shown in Table 6, and qualitatively illustrated in Fig. 9. Notice that, different from existing fully-supervised RGB-D SOD methods in designing the complicated cross-modal feature interaction strategy, our method directly exploits the learned depth semantics to promote saliency accuracy, which is simple yet effective and brand new for this field.

## 5 Failure Cases

Due to the sparsity of weak annotations, the network is usually difficult to identify the fine-grained object boundaries. As depicted in Fig. 10, although these models can effectively detect the salient objects, fine-grained details are still missing. A doable solution is to introduce auxiliary edge constraint during training. For example, the edge detection loss can be employed on the low-level features of the model, which could force the model to produce better features highlighting the object details [76, 41]. The edge maps can be generated by classical Canny operator [4] in an unsupervised manner.

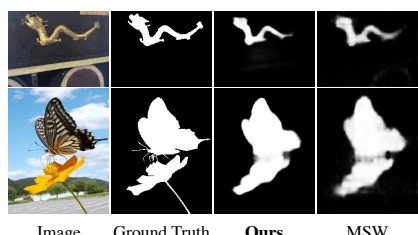

| Image | Ground Truth | **Ours** | MSW |

Figure 10: Failure cases of the existing weakly-supervised saliency methods.

Table 6: Quantitative comparison of existing fully-supervised RGB-D SOD methods. Notice that, in fully-supervised scenario, when evaluating the newly released DUTLF-Depth dataset, the specific setup used by [55] is adopted to make a fair comparison, *i.e.*, using a total of 2,985 training samples that contain 1,485 from NJUD, 700 from NLPR and 800 from DUTLF-Depth.

| Method | DUTLF-Depth [55] | | | | STERE [50] | | | | NJUD [29] | | | | NLPR [53] | | | |
|---|---|---|---|---|---|---|---|---|---|---|---|---|---|---|---|---|
| | $E_\xi$ | $F_\beta^w$ | $F_\beta$ | $\mathcal{M}$ | $E_\xi$ | $F_\beta^w$ | $F_\beta$ | $\mathcal{M}$ | $E_\xi$ | $F_\beta^w$ | $F_\beta$ | $\mathcal{M}$ | $E_\xi$ | $F_\beta^w$ | $F_\beta$ | $\mathcal{M}$ |
| CTMF $^{TCyb'17}$ [21] | .884 | .690 | .792 | .097 | .841 | .747 | .771 | .086 | .864 | .732 | .788 | .085 | .869 | .691 | .723 | .056 |
| DF $^{TIP'17}$ [58] | .842 | .542 | .748 | .145 | .691 | .596 | .742 | .141 | .818 | .552 | .744 | .151 | .838 | .524 | .682 | .099 |
| PCA $^{CVPR'18}$ [6] | .858 | .696 | .760 | .100 | .887 | .801 | .826 | .064 | .896 | .811 | .844 | .059 | .916 | .772 | .794 | .044 |
| TANet $^{TIP'19}$ [7] | .866 | .712 | .779 | .093 | .893 | .804 | .835 | .060 | .893 | .812 | .844 | .061 | .916 | .789 | .795 | .041 |
| PDNet $^{ICME'19}$ [95] | .861 | .650 | .757 | .112 | .880 | .799 | .813 | .071 | .890 | .798 | .832 | .062 | .876 | .659 | .740 | .064 |
| MMCI $^{PR'19}$ [8] | .855 | .636 | .753 | .113 | .873 | .757 | .829 | .068 | .878 | .749 | .813 | .079 | .871 | .688 | .729 | .059 |
| CPFP $^{CVPR'19}$ [86] | .814 | .644 | .736 | .099 | .912 | .808 | .830 | .051 | .895 | .837 | .850 | .053 | .924 | .820 | .822 | .036 |
| DMRA $^{ICCV'19}$ [55] | .927 | .858 | .883 | .048 | .923 | .841 | .876 | .049 | .908 | .853 | .872 | .051 | .942 | .845 | .855 | .031 |
| SSF $^{CVPR'20}$ [83] | .946 | .894 | .914 | .034 | .921 | .850 | .867 | .046 | .913 | .871 | .886 | .043 | .949 | .874 | .875 | .026 |
| A2dele $^{CVPR'20}$ [56] | .924 | .864 | .890 | .043 | .915 | .855 | .874 | .044 | .897 | .851 | .874 | .051 | .945 | .867 | .878 | .028 |
| JL-DCF $^{CVPR'20}$ [19] | .931 | .863 | .883 | .043 | .919 | .857 | .869 | .040 | - | - | - | - | .954 | .882 | .878 | **.022** |
| S2MA $^{CVPR'20}$ [42] | .921 | .861 | .866 | .044 | .907 | .825 | .855 | .051 | - | - | - | - | .938 | .852 | .853 | .030 |
| UCNet $^{CVPR'20}$ [74] | .903 | .821 | .856 | .056 | .922 | .867 | .885 | **.039** | - | - | - | - | .953 | .878 | .890 | .025 |
| PGAR $^{ECCV'20}$ [9] | .944 | .889 | .914 | .035 | .919 | .856 | .880 | .041 | .915 | .871 | .893 | .042 | .955 | .881 | .885 | .024 |
| D3Net $^{NNLS'20}$ [17] | .847 | .668 | .756 | .097 | .920 | .845 | .855 | .046 | .913 | .860 | .863 | .047 | .943 | .854 | .857 | .030 |
| CMWN $^{ECCV'20}$ [37] | .916 | .831 | .866 | .056 | .917 | .847 | .869 | .043 | .910 | .855 | .878 | .047 | .940 | .856 | .859 | .029 |
| BBSNet $^{ECCV'20}$ [18] | .833 | .663 | .774 | .120 | .925 | .858 | .885 | .041 | .924 | .884 | .902 | **.035** | .952 | .879 | .882 | .023 |
| DANet $^{ECCV'20}$ [91] | .925 | .847 | .884 | .047 | .914 | .830 | .858 | .047 | - | - | - | - | .949 | .858 | .871 | .028 |
| FRDT $^{ACMM'20}$ [84] | .941 | .878 | .902 | .039 | .925 | .858 | .872 | .042 | .917 | .862 | .879 | .048 | .946 | .863 | .868 | .029 |
| ATSA $^{ECCV'20}$ [78] | .947 | .901 | .918 | .032 | .919 | .866 | .874 | .040 | .921 | .883 | .893 | .040 | .945 | .867 | .876 | .028 |
| HDFNet $^{ECCV'20}$ [51] | .934 | .865 | .892 | .040 | .925 | .863 | .879 | .040 | .915 | .879 | .893 | .038 | .948 | .869 | .878 | .027 |
| **Ours (Fully Sup.)** | **.949** | **.908** | **.934** | **.030** | **.929** | **.876** | **.896** | **.039** | **.926** | **.885** | **.906** | .038 | **.959** | **.892** | **.905** | **.022** |

# 6 Conclusion

To tackle the new problem of weakly-supervised RGB-D salient object detection, we propose in this paper an end-to-end approach based on iterative updates of the internal pseudo-labels. This allows us to leverage depth information in eliminating non-salient background noises and generating reliable depth-refined pseudo-labels. Textual semantics is incorporated in the fill-in-the-blank fashion, which is used to estimate the confidence scores of pseudo-labels. Extensive experiments demonstrate the effectiveness and efficiency of our approach. In addition, our method is very generic and can be easily adapted to fully-supervised and unsupervised paradigms. In these scenarios, our variants also obtain superior performance over the existing state-of-the-art dedicated methods.

**Acknowledgement.** This research was supported by the University of Alberta Start-up Grant, UAHJIC Grants, and NSERC Discovery Grants (No. RGPIN-2019-04575). The authors are grateful to the anonymous reviewers for their valuable suggestions in improving the quality of the paper.

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
