# Supplementary Material: Joint Semantic Mining for Weakly Supervised RGB-D Salient Object Detection

**Jingjing Li**[1,*],  **Wei Ji**[1,*(✉)],  **Qi Bi**[2],  **Cheng Yan**[3],  **Miao Zhang**[4],
**Yongri Piao**[4],  **Huchuan Lu**[4,5],  **Li Cheng**[1]

[1]University of Alberta, Canada    [2]Wuhan University, China    [3]Tianjin University, China
[4]Dalian University of Technology, China    [5]Pengcheng Lab, Shenzhen, China

In this supplement, we first summarize the notation & definition used in this paper in Sec. 1, and give more thorough review about the recent efforts in fully-supervised RGB-D salient object detection and related RGB-based SOD approaches with low-cost annotations in Sec. 2. Then, in Sec. 3, we elaborate on the detailed network structure of our TSM, and describe the training objective for each component of the JSM framework. In Sec. 4, we provide more detailed information of the proposed *CapS* dataset. Furthermore, we give more experimental results to demonstrate the superiority of our method in Sec. 5. These results consistently indicate the reasonability and effectiveness of the proposed method. Finally, we discuss the potential limitations which can be addressed in the near future in Sec. 6.

## 1   Notation and Definition

| Notation | Definition |
|---|---|
| $\mathcal{I}$ | The training RGB image. |
| $\mathcal{D}_{map}$ | The raw depth map paired with $\mathcal{I}$. |
| $\mathcal{S}_{pred}$ | The saliency prediction produced by the saliency network. |
| $\mathcal{D}_{mask}$ | The saliency-guided depth mask generated by spatial supervision generation module in our SSM. |
| $\mathcal{D}_{\mathcal{S}}$ | The learned depth semantics generated by the depth network. |
| $k \in \{1, .., K\}$ | The image-level category label of a training sample, where $K$ is the total number of salient categories. |
| $\mathbf{c} = \{\mathbf{c}_i\}_{i=1}^{n_c}$ | The caption description of a training sample. |
| $n_c$ | The word number of the caption. |
| $\mathbf{c}_i \in \mathbb{R}^{d \times 1}$ | The word embedding of the $i$-th word in caption $\mathbf{c}$, where $d$ is its dimension. |
| $\mathcal{X}$ | The position of the salient word (category) in the caption. |
| $\hat{\mathbf{c}} \in \mathbb{R}^{d \times n_c}$ | The masked version of caption where the salient word $\mathbf{c}_{\mathcal{X}}$ in caption $\mathbf{c}$ is masked with a special symbol. |
| $\mathcal{S}_{\mathcal{D}}^{t}$ | The depth-refined pseudo-label at current training round. |
| $\mathcal{S}_{map}^{t}$ | The pseudo-label at current training round. |
| $\mathcal{S}_{map}^{t+1}$ | The updated pseudo-label via the JSM at the end of current training round. |
| $\mathcal{S}_{GT}$ | The pixel-level ground-truth label. |
| $F_{vis}$ | The visual features extracted by the saliency network. |
| $\lambda_d$ | The hyper-parameter in the SSM, to control the degree of the subtracted background noises. |
| $\tau$ | The update interval in the JSM, *i.e.,* the granularity of label updating. |
| $\mathbf{e}_{\mathcal{X}} \in \mathbb{R}^{K \times 1}$ | The energy vector of the masked salient word produced by transformer-like net. |
| $\mathcal{SC}$ | The confidence score produced by the TSM, based on saliency-filtered visual feature (*i.e.,* different pseudo-labels). |

35th Conference on Neural Information Processing Systems (NeurIPS 2021)

## 2 Related Work

### 2.1 Fully-supervised RGB-D Salient Object Detection

Although many works [22, 21, 24, 50, 38, 9, 44] have devoted to RGB-based salient object detection (SOD) and have achieved appealing performance, they might fail when coping with complex scenarios, such as cluttered background and low-intensity environment. This naturally leads to the incorporation of depth information in addition to the conventional RGB image as input, known as RGB-D SOD. [31, 4, 15, 14] demonstrate that depth information, containing spatial structure and 3D layout cues in a scene, is helpful to alleviate the challenging scenarios.

With explicit pixel-level supervisions, existing RGB-D SOD methods mainly concentrate on learning multi-modal feature representations, by designing feature fusion strategies to promote the interactions between visual features from RGB image and complementary spatial features from depth map. Chen *et al.* [11, 6] employ a two-stream CNNs-based model and perform fusion by adding or concatenating paired features at shallow or deep layers. In [4], they design a fusion network, where cross-level features are progressively combined. To further promote multi-modal feature interactions, Liu *et al.* [23] utilize a residual fusion module to integrate depth cues into RGB stream, and exploit self-attention and mutual attention to capture the contexts of fused features. Li *et al.* [19] design a cross-modal weighting strategy to encourage comprehensive interactions between RGB and depth information. We refer interested researchers to recent comprehensive surveys [52, 36, 2, 49, 9, 17, 48, 47] that have well studied fully-supervised SOD field.

However, these methods often require costly pixel-level annotations, which are tedious and time-consuming to obtain. This motivates us to consider a weakly-supervised approach. To our best knowledge, it is the first effort to address weakly-supervised RGB-D SOD problem, *i.e.,* only image-level labels are available. In what follows, our focus will be mainly toward related weakly-supervised methods, where the differences of our approach from existing methods would be clarified.

### 2.2 Salient Object Detection with Low-cost Annotations

#### 2.2.1 Image-level Supervision

To avoid requiring laborious per-pixel labels, some methods attempt to learn saliency from low-cost image-level supervisions, such as image tags (or categories), and image captions. Wang *et al.* [35] introduce a foreground inference network with object category labels for learning salient object detector, which requires less annotation efforts. Hsu *et al.* [12] design a category-driven map generator to learn saliency from class activation map. Li *et al.* [20] develop a graphical model combined with CNNs to perform model updating, which corrects the ambiguity of noisy labels. Due to the limited information provided by image-level tags, the trained networks usually highlight only the most discriminative regions, which makes it difficult to detect the whole object. Zeng *et al.* [42] use image captions that describe the main content of an image, to provide more comprehensive cues to complement image category. They utilize the pseudo-labels from classification network and caption generation network to jointly train the target saliency models.

#### 2.2.2 Sparse Pixel-level Supervision

Recent works [41, 45] attempt to explore other weak supervision signal, *i.e,* scribble annotation. It only annotates a small set of image pixels as foreground or background annotations, which is low-cost. However, due to the annotation sparsity, object structure and details cannot be easily inferred. Zhang *et al.* [45] introduce a gated structure-aware loss function as well as an auxiliary edge detection network to enhance the complete structure of foreground object. Meanwhile, Yu *et al.* [41] explore self-consistency of multi-scale outputs and design a local coherence loss to propagate the labels to unlabeled regions based on image features. This allows the model to detect smoother and integral salient objects.

#### 2.2.3 Free-cost Supervision

Compared to accurate pixel-level or low-cost supervision signals, SOD with free-cost supervision (or unsupervised SOD) that do not rely on such annotations is naturally considered. It is generally categorized into handcrafted methods and deep unsupervised SOD with noisy labels. For the first

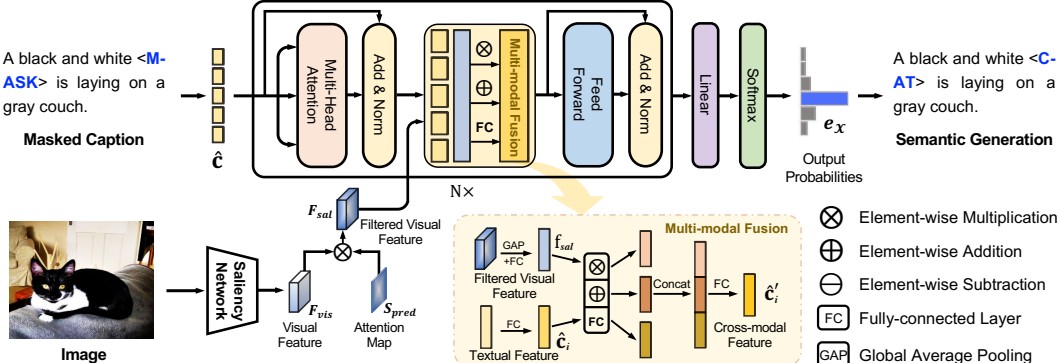

Figure 1: The detailed structure of our textual semantic modeling (TSM), during training stage.

class, handcrafted methods are mainly based on the manually-crafted human priors, including depth cues [29, 7], global priors [32], center priors [53] and contrast priors [33]. Secondly, building upon the powerful learning capacity of CNN, deep unsupervised SOD methods achieve appealing performance over traditional methods. They usually use the noisy output produced by traditional methods as pseudo-label for training saliency network. Zhang *et al.* [43] define a fusion strategy to combine the pseudo-labels from handcrafted methods on super-pixel and image-level. In [46], a noise modeling is proposed to fit the noise distribution of pseudo-label. Rather than the direct use of noisy pseudo-labels, Nguyen *et al.* [26] refine pseudo-label iteratively via self-supervision technique, and achieve better performance. Notice that, in this paper, our variant with SSM can be adapted to unsupervised setting, which is free-cost for human annotations.

## 2.3 Weakly-supervised RGB-D Salient Object Detection

In this work, we systematically formulate a new problem on *weakly-supervised RGB-D salient object detection*, and tackle its new challenges. (**1**) Considering the large variations in the raw depth map and the lack of explicit pixel-level supervisions, our SSM is designed to capture the saliency-specific depth semantics, to eliminate the background noises in the coarse saliency prediction, and to generate a depth-refined pseudo-label. (**2**) To mitigate the noisy issue of weak supervisions, our JSM is proposed to provide internal pixel-level supervision signals, which is progressively updated by reconciling the multimodal input signals and the current information flow of the neural net. Meanwhile, a TSM is introduced to estimate the confidence scores of competing pseudo-labels, from a new perspective.

## 3 Model Details

### 3.1 Detailed Structure of Textual Semantic Modeling

Previously, the mainstream use of weak labels is to train a classification network or a caption generation network, where the by-product attention maps or Class Activation Maps [51] are leveraged to determine the potential salient regions [42, 35]. It is very different in our textual semantic modeling (TSM), where the main focus is to leverage side information (*i.e.*, image-level tags and captions) to facilitate the production of reliable training signals. Inspired by the recent success of masked language models [8], captions with missing keywords are used as input, with the expectation of the complete text being reconstructed as output. In the proposed TSM, innovatively taking as input partial text with salient word being masked, as well as the saliency-filtered visual features, our TSM is to output the reconstructed text in a fill-in-the-blank manner and to estimate the confidence scores of competing pseudo-labels.

Formally, for each training data $\mathcal{I}$, the weak labels contain caption description $\mathbf{c} = \{\mathbf{c}_i\}_{i=1}^{n_c}$, image-level category $k \in \{1, .., K\}$, and the position $\mathcal{X}$ of the salient word (category) in the caption, where $n_c$ is the word number of the caption. Let $\mathbf{c}_i \in \mathbb{R}^{d \times 1}$ be the word embedding of the $i$-th word in the caption, $K$ the total number of salient categories, and $\mathcal{X}$ an integral number. As presented in Fig. 1, the input is a masked version of caption $\hat{\mathbf{c}} \in \mathbb{R}^{d \times n_c}$ where the salient word $\mathbf{c}_{\mathcal{X}}$ in caption $\mathbf{c}$ is masked with a special symbol. In order to reconstruct the masked salient word, we filter the visual feature $F_{vis}$ from the saliency network by multiplying it with the learned saliency attention (*i.e.,*

$\mathcal{S}_{pred}$). We then obtain the saliency-filtered visual feature $F_{sal}$, and transform it to a feature vector $\mathbf{f}_{sal} \in \mathbb{R}^{d \times 1}$ for subsequent cross-modal fusion using a GAP (global average pooling) operation and a fully-connected convolution layer.

The center component of the TSM module is a transformer-like encoder, composed of a stack of layers: a multi-head self-attention sub-layer, a multi-modal fusion sub-layer, and a fully-connected (FC) feed-forward sub-layer. The structures of the first and third sub-layers are similar to the original Transformer [34]. In the multi-modal fusion sub-layer, we use three parallel operations to promote sufficient cross-modal feature interactions: element-wise multiplication $\otimes$, element-wise addition $\oplus$, and concatenation (denoted by $\|$) followed by FC. Then three outputs are concatenated and followed by a FC to change the feature dimension. Note that this fusion operation is performed word-wise, as

$$\hat{\mathbf{c}}_i' = \text{FC}\left((\mathbf{f}_{sal} \otimes \hat{\mathbf{c}}_i) \| (\mathbf{f}_{sal} \oplus \hat{\mathbf{c}}_i) \| \text{FC}(\mathbf{f}_{sal} \| \hat{\mathbf{c}}_i)\right). \tag{1}$$

Collectively, through the textual encoder, the cross-modal representation is in the following form,

$$\hat{\mathbf{c}}' = \mathbf{Enc}(\hat{\mathbf{c}}, \mathbf{f}_{sal}). \tag{2}$$

To predict the masked salient word, the energy vector $\mathbf{e}_{\mathcal{X}} \in \mathbb{R}^{K \times 1}$ is computed over all categories by a fully-connected layer and softmax function $\sigma(\cdot)$, as

$$\mathbf{e}_{\mathcal{X}} = \sigma(W_e \hat{\mathbf{c}}_{\mathcal{X}}' + b_e). \tag{3}$$

Here $W_e$ and $b_e$ are training parameters of the FC layer. Therefore, we obtain $\mathbf{e}_{\mathcal{X}}[k]$, the output probability of salient category $k$. Finally, the training loss for the proposed TSM is:

$$\mathcal{L}_{tsm} = \frac{1}{N} \sum_{n=1}^{N} (-\log(\mathbf{e}_{\mathcal{X}_n}^n[k_n])). \tag{4}$$

where $N$ is the number of training samples in each mini-batch. Once trained, the TSM module is then use to estimate the confidence scores of pseudo-labels when performing pseudo-label update.

## 3.2   Training Objective

In the training stage, the saliency network, spatial semantic modeling (SSM) and textual semantic modeling (TSM) are trained simultaneously, without back-propagating gradients to each other. We use the standard binary cross entropy loss to train the saliency network, as in

$$\mathcal{L}_{sal} = \frac{1}{N} \sum_{n=1}^{N} (-\mathcal{S}_{map}^n \cdot \log(\mathcal{S}_{pred}^n) - (1 - \mathcal{S}_{map}^n) \cdot \log(1 - \mathcal{S}_{pred}^n)), \tag{5}$$

where $\mathcal{S}_{map}$ is the current pseudo-label, and $\mathcal{S}_{pred}$ means the prediction from saliency network. Meanwhile, we employ the mean square error (*i.e.,* MSE) loss between saliency-guided depth mask $\mathcal{D}_{mask}$ and depth semantic prediction $\mathcal{D}_{\mathcal{S}}$ to train the SSM, as in

$$\mathcal{L}_{ssm} = \frac{1}{N} \sum_{n=1}^{N} (\mathcal{D}_{mask}^n - \mathcal{D}_{\mathcal{S}}^n)^2. \tag{6}$$

In terms of the TSM, we use $\mathcal{L}_{tsm}$ of the Eq. 4 to update its training parameters.

Furthermore, the pseudo-label update operation using our joint semantic mining is iteratively conducted every five training epochs, described in Sec. 3.4 of the main text. This achieves trustworthy supervision signal to train the target saliency network.

## 4   The CapS Dataset

Fig. 2 presents the statistics and examples of our *CapS* dataset, where various image-level supervisions are provided to augment the existing RGB-D SOD training dataset. Specifically, compared to original RGB-D SOD benchmark with the annotated pixel-level supervisions, we further provide the image-level categories and captions in the *CapS*. We summarize the 23 super-categories containing 100 salient categories tailored for SOD task. Each data corresponds to a salient category, which is included in the caption. In terms of multiple objects with different categories in an image (1.1% cases), we

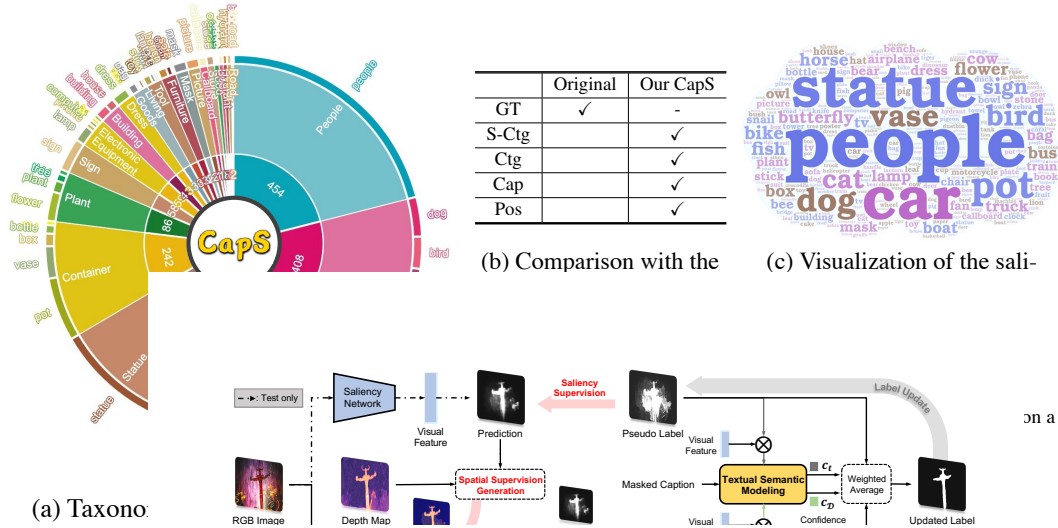

|  | Original | Our CapS |
|---|---|---|
| GT | ✓ | - |
| S-Ctg |  | ✓ |
| Ctg |  | ✓ |
| Cap |  | ✓ |
| Pos |  | ✓ |

(b) Comparison with the

(c) Visualization of the sali-

(a) Taxono...

Figure 2: Statistics and examples of the introduced *CapS* dataset. (a) Taxonomic system and pie chart distribution. (b) The provided annotations. GT: ground-truth; Cap: caption; S-Ctg: super-category; Ctg: category; Pos: the position of salient word in the caption. (c) Word cloud distribution. (d) An example of various annotations.

select the dominated object as salient category, which is discussed in Sec. 6. Next, the position of the salient word (category) in each of the captions is also given in a semi-automatic manner, as described in Sec. 3.5 of the main text.

The numerical statistics of our *CapS* is listed in Table 1. *Our CapS dataset is publicly available at* https://github.com/jiwei0921/JSM. Hopefully this could encourage more contributions to this community.

Table 1: Numerical statistics of the introduced *CapS* dataset, containing 23 super-categories with 100 categories on image-level annotations.

| Animal | | | | | | | | | | | | | | |
|---|---|---|---|---|---|---|---|---|---|---|---|---|---|---|
| owl | dog | chicken | bird | horse | fish | pigeon | butterfly | cat | zebra | bear | cow | duck | bee | tortoise |
| 8 | 82 | 3 | 59 | 37 | 21 | 4 | 51 | 57 | 4 | 17 | 17 | 4 | 10 | 2 |

| | | | | | Animal | | | | | | | Ball | Book | Building |
|---|---|---|---|---|---|---|---|---|---|---|---|---|---|---|
| dinosaur | monkey | lion | snail | deer | giraffe | leopard | tiger | pig | crocodile | panda | basketball | football | book | house |
| 5 | 2 | 2 | 13 | 2 | 1 | 2 | 2 | 1 | 1 | 1 | 1 | 1 | 7 | 15 |

| Building | | Callboard | | Container | | | | | | | | | Dress | |
|---|---|---|---|---|---|---|---|---|---|---|---|---|---|---|
| building | tower | bridge | callboard | vase | pot | bowl | bottle | box | dustbin | cup | plate | dress | towel | shoes |
| 21 | 5 | 2 | 13 | 69 | 118 | 7 | 14 | 21 | 5 | 2 | 6 | 26 | 3 | 3 |

| Dress | | Electronic equipment | | | | | | | Food | | | | Furniture | |
|---|---|---|---|---|---|---|---|---|---|---|---|---|---|---|
| hat | suit | lamp | phone | computer | tv | machine | monitor | telescope | orange | fruit | apple | cake | bench | sofa |
| 4 | 1 | 37 | 6 | 5 | 5 | 4 | 1 | 1 | 3 | 2 | 2 | 1 | 14 | 3 |

| Furniture | | Hydrant | | Living goods | | | | | | Mask | People | Picture | Plant | |
|---|---|---|---|---|---|---|---|---|---|---|---|---|---|---|
| door | chair | window | hydrant | paper | bag | pillow | piano | clock | toy | mask | people | picture | plant | tree |
| 3 | 8 | 2 | 7 | 2 | 13 | 3 | 1 | 6 | 5 | 22 | 454 | 21 | 19 | 8 |

| Plant | | | | Poster | Road | Sign | Statue | Stone | Tool | | | | | |
|---|---|---|---|---|---|---|---|---|---|---|---|---|---|---|
| flower | bush | leaf | coral | poster | road | sign | statue | stone | stick | fan | knife | lantern | wheel | handle |
| 48 | 4 | 4 | 3 | 7 | 2 | 58 | 256 | 12 | 12 | 6 | 4 | 4 | 4 | 2 |

| Vehicle | | | | | | | | | | Super-category: 23 | | | | |
|---|---|---|---|---|---|---|---|---|---|---|---|---|---|---|
| truck | car | train | airplane | motorcycle | boat | bus | bike | helicopter | tank | Category: 100 | | | | |
| 34 | 154 | 19 | 41 | 23 | 30 | 16 | 23 | 8 | 1 | Total Number: 2185 | | | | |

# 5 More Experimental Results

In this section, we visually show more experimental results of our method, in both weakly-supervised and fully-supervised settings.

**Weakly-supervised setting.** As shown in Fig. 3, our method can better capture salient regions in a scene than others, and our saliency maps are closest to the ground-truth label. This benefits from the proposed joint semantic mining framework that provides trustworthy supervisory signals to train the saliency network.

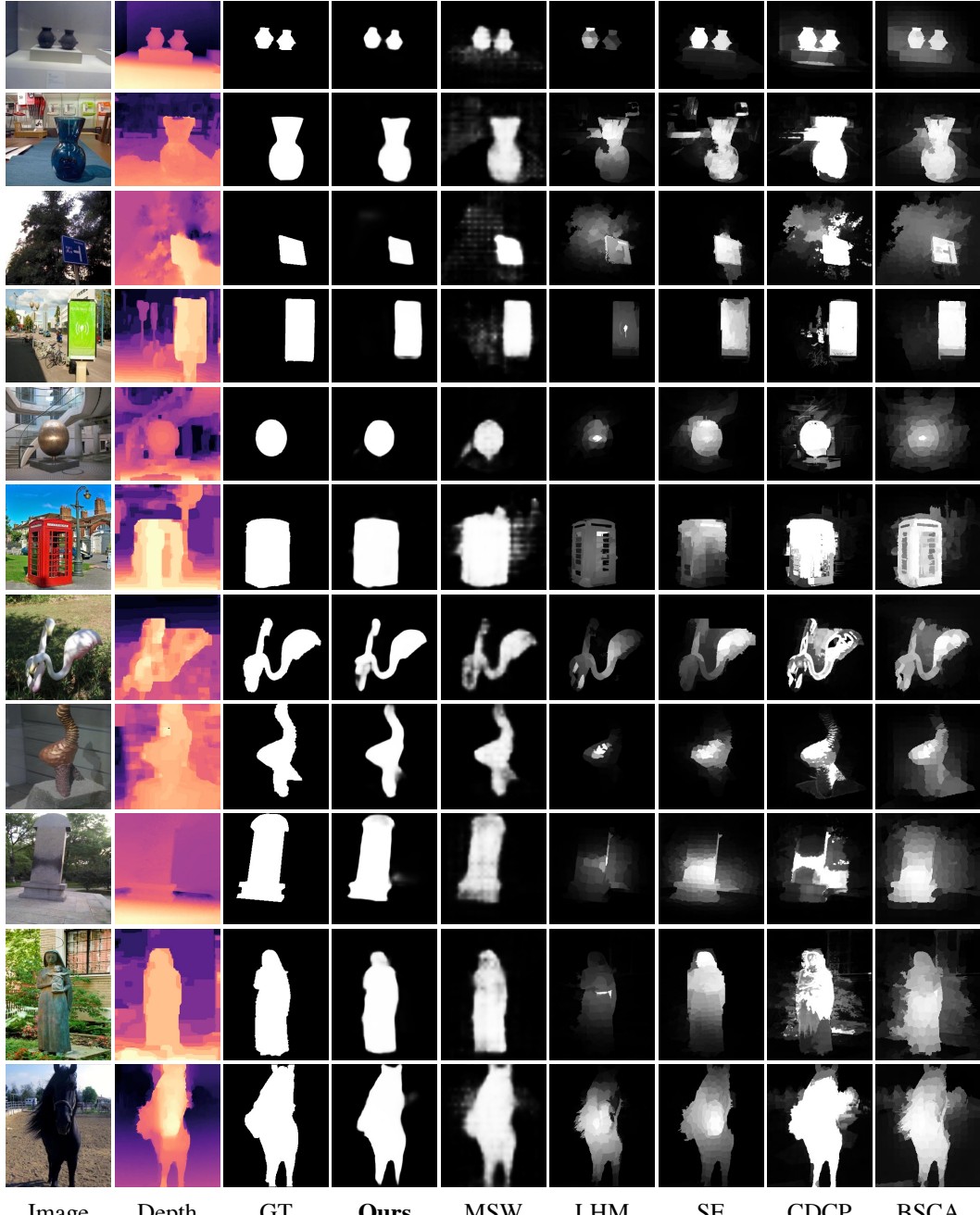

| Image | Depth | GT | **Ours** | MSW | LHM | SE | CDCP | BSCA |

Figure 3: Visual comparisons of weakly-supervised and unsupervised saliency models. 'GT' represents the ground-truth saliency for reference only.

**Fully-supervised setting.** We show the visual results of our fully-supervised variant and several top-ranking RGB-D models in Fig. 4. It is observed that our fully-supervised variant produces better saliency predictions. In addition, we further apply our SSM to several existing RGB-D salient object detection methods, to verify the scalability of our method. Specifically, the learned depth semantics from the

Table 2: Application to existing RGB-D SOD methods.

| $*$ | TANet [5] | | CTMF [11] | | PCA [4] | | MMCI [6] | | CMWN [19] | |
|---|---|---|---|---|---|---|---|---|---|---|
| | Ori | **Our** | Ori | **Our** | Ori | **Our** | Ori | **Our** | Ori | **Our** |
| $E_\xi$ | .916 | .938 | .869 | .918 | .916 | .929 | .871 | .910 | .940 | .951 |
| $F_\beta^w$ | .789 | .822 | .691 | .752 | .772 | .799 | .688 | .753 | .856 | .879 |
| $F_\beta$ | .795 | .848 | .723 | .794 | .794 | .836 | .729 | .789 | .859 | .885 |
| $\mathcal{M}$ | .041 | .032 | .056 | .044 | .044 | .039 | .059 | .045 | .029 | .023 |

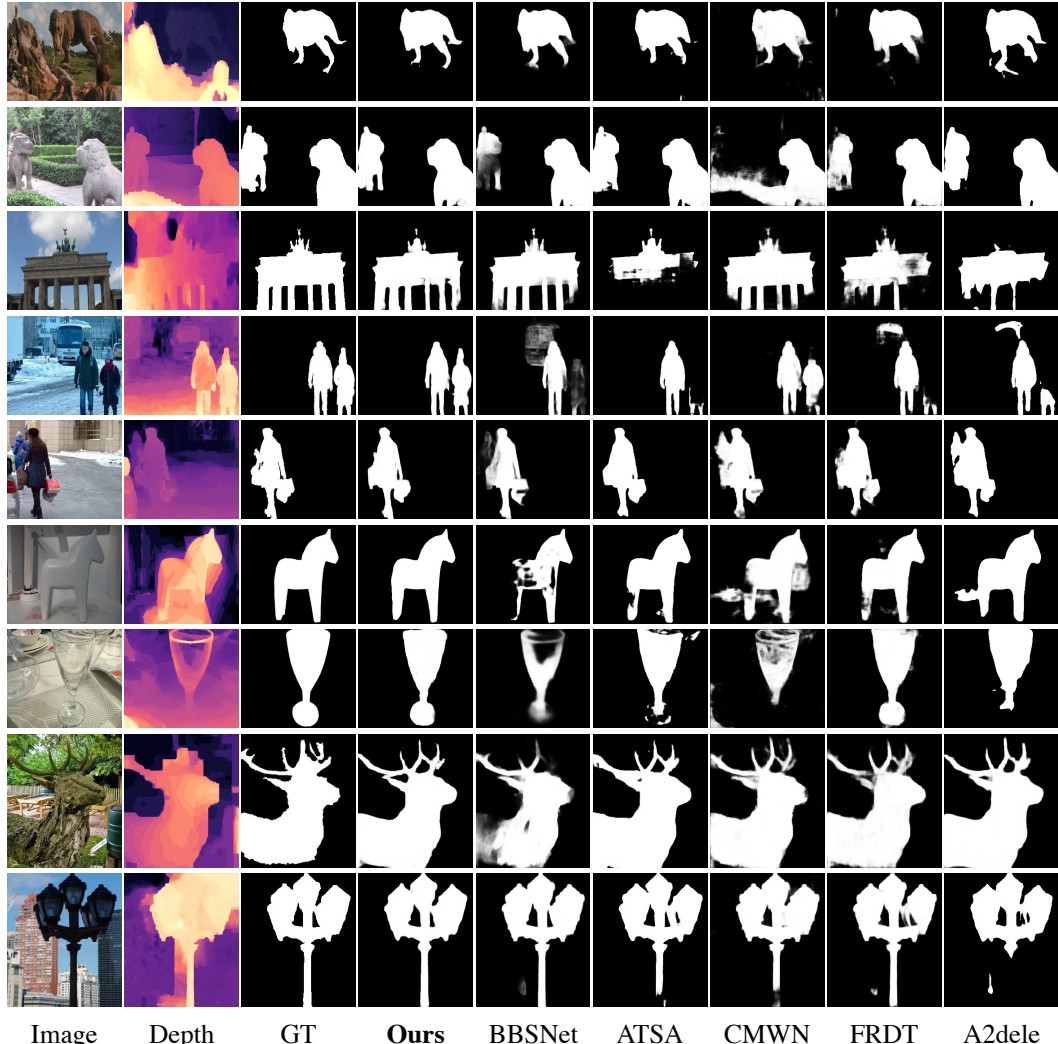

| Image | Depth | GT | **Ours** | BBSNet | ATSA | CMWN | FRDT | A2dele |

Figure 4: Visual comparisons of top-ranked RGB-D saliency models under the fully-supervised setting. 'GT' means the ground-truth saliency.

depth network and the saliency prediction from various models (*e.g.*, CTMF, PCA) are fed into the background noise suppression block in SSM, which obtains improved saliency. Both the original results of these methods and the new results of incorporating our SSM (denoted as Ori *vs.* Our) on the NLPR benchmark are reported in Table 2. These results consistently demonstrate the generic applicability and superiority of our method.

**Performance on RGB benchmark.** Benefiting from our additional merit, *i.e.*, not relying on depth during inference, we also test our weakly-supervised model on popular RGB-based SOD benchmark. To be specific, we use our pretrained model to test on the popular RGB benchmark DUT-OMRON [40]. For MSW [42], the saliency maps provided by the authors are used for evaluation. The results are as follows (MSW / Ours): 0.763 / 0.786 on $E_\xi$, 0.527 / 0.563 on $F_\beta^w$, 0.609 / 0.633 on $F_\beta$, 0.114 / 0.093 on MAE metric.

Table 3: Analysis of the long-tailed problem in RGB-D SOD.

| Model Setup | NJUD [18] | | NLPR [29] | | STERE [28] | | DUT-D [30] | |
|---|---|---|---|---|---|---|---|---|
| | $F_\beta$ | $\mathcal{M}$ | $F_\beta$ | $\mathcal{M}$ | $F_\beta$ | $\mathcal{M}$ | $F_\beta$ | $\mathcal{M}$ |
| Our JSM | .717 | .133 | .770 | .060 | .778 | .095 | .797 | .093 |
| Our JSM with Re-sampling | .728 | .129 | .781 | .057 | .792 | .091 | .803 | .092 |

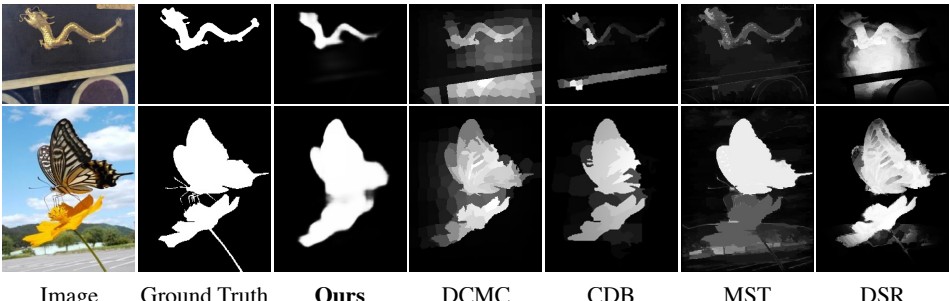

| Image | Ground Truth | **Ours** | DCMC | CDB | MST | DSR |

Figure 5: Failure cases of the existing weakly-supervised and unsupervised saliency methods.

## 6 Discussion and Outlook

In this section, we summarize three potential research problems on *weakly-supervised RGB-D salient object detection*. Meanwhile, the feasible solutions are given for reference. Hopefully this could encourage more inspirations and contributions to this community and further pave the way for its booming future. They are summarized as follows:

**(1) Fine-grained problem.** Due to the sparsity of weak annotations, the network is usually difficult to identify the fine-grained object boundaries. As depicted in Fig. 5, although these models can effectively detect the salient objects, the fine-grained details are missing. A doable solution is to introduce auxiliary edge constraint. For example, the edge detection loss is employed to low-level features of model, which forces model to produce the features highlighting object details [45, 21, 50]. The edge maps can be generated by classical Canny operator [3].

**(2) Long-tailed problem.** In natural image field, a long-tailed distribution of category frequency in the large dataset is ubiquitous and inevitable. As shown in Table 1, existing RGB-D saliency training set contains some rare categories. To address this problem, a widely-used re-sampling method [10] is adopted in our JSM. It is shown in Table 3 that our method consistently achieves performance improvements on four benchmark datasets.

**(3) Multi-label problem.** In terms of multi-label problem in the TSM, *i.e.,* an image consists of multiple salient objects with different categories, one straightforward solution is to translate it as a single-label & multi-class task as in this work. Another way is to average the results of multiple categories as the final score. Experiments indicates that the two ways have slight performance difference ($\Delta = 0.0015$ in terms of average MAE error over four benchmarks), because statistically there exists only $1.1\%$ multi-label cases in the existing RGB-D dataset. Besides, some interesting ideas can be further explored, such as object rank [25] and model ensemble [37, 13].

In the future, we are planning to introduce more weak annotations in our *CapS* dataset, such as bounding box, pixel-level scribble annotations. We will also apply our method to other fields, *e.g.*, medical analysis [1, 16], semantic segmentation [27], object recognition [39].