# OpenReview forum: "Joint Semantic Mining for Weakly Supervised RGB-D Salient Object Detection"
_NeurIPS.cc/2021/Conference — NeurIPS 2021 Poster_

### Official Review · Reviewer_n5it · 2021-07-11

**Rating:** 5
**Confidence:** 4

**Summary:**

This article addresses salient object detection using depth maps and image captions as supervised signals. The proposed network includes a SSM for spatial semantic modelling and a TSM for cross-modal semantic mining. Additionally, a CapS dataset is built to extend existing datasets with extra weak annotations.

**Limitations And Societal Impact:**

Yes. The supplement has provided some discussions of the limitations of the work.

**Main Review:**

PROS:
- The paper is well written and easy to follow, with clear, complete and concise explanations.
- The paper provides a good and detailed description of the proposed network.
- The ablation study is comprehensive.

CONS:
- My major concern goes to the design of network architecture, especially the necessity of the depth network. How about using $D_\text{mask}$ directly for $D_s$? Please clarify the motivation to predict a depth map rather than directly using the true depth map?
- The approach exploits a combination of multiple sources (\ie, depth maps, captions, tags)，which leads to unfair comparisons with some leading methods that only rely on one or two types of knowledge. This make the results somewhat unconvinced.
- L339 claims the model to be end-to-end learnable, however, the modules are "independently trained" (L207). I am guessing whether the network can be end-to-end trained in practise. It will be good to study the performance difference.
- Additionally, is CRF applied during every update? How is it implemented to be end-to-end trainable?
- How does the proposed model perform in the tailed categories? The long-tail problem seems to be obvious in CapS (as shown in Fig. 2 in the supplement)
- It is not that obvious why does the model achieves the best results in the fully supervised setting, since the model will degrade to a simple two-branch neural network. More discussions are preferred.
- Only RGB-D datasets are taken into account in the experiments. What's the reason for this, especially considering that the network does not rely on depth information during inference? What's the model performance on RGB only benchmarks?

**Time Spent Reviewing:**

4

---

> ### Author Response · Authors · 2021-08-10
> **Response to Reviewer n5it [Part 1 /2]**
>
> **We appreciate the time and effort you spent reviewing our paper. In the following response, we will clear some misunderstanding through the rebuttal and address the questions point-by-point.**
>
> --------------
>
> ***Question1**) My major concern goes to the design of network architecture, especially the necessity of the depth network. How about using $D_{mask}$ directly for $D_s$? Please clarify the motivation to predict a depth map rather than directly using the true depth map?*
>
> **Answer1**: Thanks for the comments. We claimed the motivation of our network design for not directly using true depth map in lines 35-40 in the manuscript.
> An intuitive example is also given to further explain the claim. In Fig. 1, similar depth values are shared by the foreground cat and the underneath couch, making it difficult to discern the salient object from backgrounds. Thus, a direct adoption of the true depth map may not necessarily translate to a good result.
> To tackle this challenge, our SSM module is engaged in learning saliency-specific depth.
> **Three aspects have demonstrated the necessity of our depth network.**
> **First, Table 1 quantitatively verified the effectiveness of our depth network**, which shows that using our depth semantics $D_S$ from the depth network is able to effectively suppress the noise in the pseudo-labels, leading to a significant amount of 10.3% error reduction on average (ablation models in Fig. 5 (a) vs. (b) and numerical results in Table 1).
> **Second, we also qualitatively presented the $D_{mask}$ and the $D_s$ in Fig. 4.**  Compared to $D_s$, $D_{mask}$ usually exhibits large value variations (the 5$^{th}$ and 6$^{th}$ columns in Fig. 4).
> **Finally, for “how about using $D_{mask}$ directly for $D_s$”, we conducted experiments as listed in Tab.A below.**
> Degraded performance is observed when replacing the $D_s$ with $D_{mask}$ in our SSM.
> **These results consistently demonstrate the rationality and effectiveness of our approach.**
>
> |            [**Tab.A**]            |             NJUD            |             NLPR            |            STERE            |
> |:----------------------------:|:---------------------------:|:---------------------------:|:---------------------------:|
> |               -              | $F_{\beta}$ & $\mathcal{M}$ | $F_{\beta}$ & $\mathcal{M}$ | $F_{\beta}$ & $\mathcal{M}$ |
> |      SSM with $D_{mask}$     |         .665 & .158         |         .642 & .090         |         .703 & .125         |
> | SSM with $D_{s}$ (i.e., **Our design**) |         .689 & .147         |         .698 & .073         |         .738 & .111         |
>
> --------------
>
> ***Question2**) The approach exploits a combination of multiple sources (i.e., depth maps, captions, tags), which leads to unfair comparisons with some leading methods that only rely on one or two types of knowledge. This make the results somewhat unconvinced.*
>
> **Answer2**: We would like to clarify that our work aims at tackling the RGB-D salient object detection (RGB-D SOD) with cheap weak supervisions.
> To our best knowledge, this is **the first such attempt in RGB-D SOD**.
> It can greatly save human efforts compared to fully-supervised RGB-D SOD.
> **Since there is no weakly-supervised RGB-D SOD method available**, we additionally provided the performance of two RGB-based weakly-supervised methods, WSS and MSW, as described in lines 295-298.
> **Such comparisons are not meant to show the superiority of our method against those methods but only provide  observational evidence for the related works.**
> In addition, **to verify the effectiveness of the proposed method, we have conducted extensive ablation analyses on weakly-supervised setting and adaptation experiments on both fully-supervised and unsupervised settings** (Figs. 4, 7, 8 & Tables 1, 2, 5 of the manuscript and supplementary materials).
> In this paper, we clearly marked the methods under RGB setting or RGB-D setting using different notations (Table 3 and Table 4).
>
> --------------
>
> ***Question3**) L339 claims the model to be end-to-end learnable, however, the modules are "independently trained" (L207). I am guessing whether the network can be end-to-end trained in practise. It will be good to study the performance difference. Additionally, is CRF applied during every update? How is it implemented to be end-to-end trainable?*
>
> **Answer3**: In this paper, the `independently trained' statement in line 207 means that the gradients in SSM and TSM do not back-propagate to each other during training.
> For the use of CRF, as described in lines 208-217, at the end of each training round, the CRF (i.e., off-the-shelf pydensecrf python package) is directly applied to $S_{map}^{t+1}$, which then could provide more trustworthy signals for model training.
> **We will release the source code & dataset.** This facilitates interested researchers reproducing our results.
>
> --------------
>
>
> ***Question4**) How does the proposed model perform in the tailed categories? The long-tail problem seems to be obvious in CapS (as shown in Fig. 2 in the supplement).*
>
> **Answer4**: Long-tail problem has always been an universal and inevitable problem for a variety of tasks in the computer vision community. In the supplement, we employed popular technique (i.e., re-sampling) to alleviate this problem, as shown in the supplement.
> Following the suggestion, we present in Tab.B the performance of the proposed model in the long-tailed categories.
> Specifically, we collect the long-tailed training examples where the number of examples in their categories are lower than $N$ (denoted by @N). We test three different values, $N\in\{15, 10, 5\}$.
> Their pseudo-labels are compared with the ground-truth labels to calculate the E-measure, weighted F-measure, F-measure, and mean absolute error scores.
> '*Full data*' represents the overall performance of all training examples.
>
>
> |                     [**Tab.B**]   *Full data*               	|                           @15                              	|                            @10                             	|                            @5                             	|
> |:---------------------------------------------------------:	|:---------------------------------------------------------:	|:---------------------------------------------------------:	|:---------------------------------------------------------:	|
> | $E_{\xi}$ / $F_{\beta}^{w}$ / $F_{\beta}$ / $\mathcal{M}$ 	| $E_{\xi}$ / $F_{\beta}^{w}$ / $F_{\beta}$ / $\mathcal{M}$ 	| $E_{\xi}$ / $F_{\beta}^{w}$ / $F_{\beta}$ / $\mathcal{M}$ 	| $E_{\xi}$ / $F_{\beta}^{w}$ / $F_{\beta}$ / $\mathcal{M}$ 	|
> |                [.770 / .624 / .695 / .118]                	|                [.761 / .616 / .690 / .125]                	|                [.750 / .603 / .677 / .130]                	|                [.746 / .597 / .670 / .133]                	|
>
>
> --------------
>
>
>
> ***Question5**) It is not that obvious why does the model achieves the best results in the fully supervised setting, since the model will degrade to a simple two-branch neural network. More discussions are preferred.*
>
> **Answer5**: **We analyze the model from the perspective of model design and experimental results.**
> Different from existing fully-supervised RGB-D SOD methods that usually design cross-modal fusion strategies (i.e., convolution or skip-connection operations) to integrate RGB and complementary depth features, **we innovatively exploit depth map to construct the spatial supervision signal; the learned depth semantics is then utilized to effectively suppress the background noise in the saliency prediction.**
> The ablation results in Table 5 have demonstrated the effectiveness of our model design in the fully supervised setting.
> Following your suggestion, **we further apply our SSM to several existing RGB-D SOD methods, to verify the scalability of our method.**
> Specifically, the learned depth semantics from the depth network and the saliency prediction from various models (e.g., CTMF, PCA) are fed into the background noise suppression block in SSM, which obtains improved saliency.
> Both the original results of these methods and the new results of incorporating our SSM (denoted as Ori vs. Our) on the NLPR benchmark are reported in Tab.C below.
> This again verifies the effectiveness of our approach.
> The analyses and results will be included in the main text. Thanks for your valuable suggestion.
>
>
>
> |   [**Tab.C**]     | [S2MA |  ---] | [CWMN |  ---] | [BBSNet |  ---] | [CPFP |  ---] | [TANet |  ---] | [MMCI |  ---] |  [PCA |  ---] | [CTMF |  ---] |
> |:---------------:|:------:|:----:|:----:|:----:|:----:|:----:|:----:|:----:|:-----:|:----:|:----:|:----:|:----:|:----:|:----:|:----:|
> |        -        |   [Ori  |  **Our**] |  [Ori |   **Our**] |  [Ori |   **Our**] |  [Ori |   **Our**] |  [Ori  |   **Our**] |  [Ori |   **Our**] |  [Ori |   **Our**] |  [Ori |   **Our**] |
> |    $E_{\xi}$    | [.938 | .950] | [.940 | .951] | [.952  | .961] | [.924 | .937] |  [.916 | .938] | [.871 | .910] | [.916 | .929] | [.869 | .918] |
> | $F_{\beta}^{w}$ | [.852 | .873] | [.856 | .879] | [.879  | .897] | [.820 | .841] |  [.789 | .822] | [.688 | .753] | [.772 | .799] | [.691 | .752] |
> |   $F_{\beta}$   | [.853 | .879] | [.859 | .885] | [.882  | .903] | [.822 | .853] |  [.795 | .848] | [.729 | .789] | [.794 | .836] | [.723 | .794] |
> |  $\mathcal{M}$  | [.030 | .025] | [.029 | .023] | [.023  | .021] | [.036 | .029] |  [.041 | .032] | [.059 | .045] | [.044 | .039] | [.056 | .044] |
>
>
> --------------

---

> > ### Author Response · Authors · 2021-08-10
> > **Response to Reviewer n5it [Part 2 /2]**
> >
> >
> > ***Question6**) Only RGB-D datasets are taken into account in the experiments. What's the reason for this, especially considering that the network does not rely on depth information during inference? What's the model performance on RGB only benchmarks?*
> >
> > **Answer6**: Since our approach is specifically designed for *RGB-D based saliency detection using cheap weak supervisions*, both the RGB and depth data are necessary during model training.
> > **Naturally, we follow the common practice of the recent fully-supervised RGB-D SOD methods**  and evaluate the performance of our approach on the widely-used RGB-D benchmarks.
> > Second, in the test stage, our model has additional benefits of not introducing extra test-time cost and not relying on the depth map.
> > Following your suggestion, **the evaluation results on three popular RGB benchmarks are provided in Tab.D below.**
> > **We will also release the source code, pre-trained model and ReadMe.txt,**  which allows researchers to test our model on any RGB or RGB-D datasets.
> >
> >
> >
> > |[**Tab.D**]|                         DUT-OMRON                         	|                           ECSSD                           	|                          PASCAL-S                         	|
> > |:--------:	|:---------------------------------------------------------:	|:---------------------------------------------------------:	|:---------------------------------------------------------:	|
> > |  Metrics 	| $E_{\xi}$ / $F_{\beta}^{w}$ / $F_{\beta}$ / $\mathcal{M}$ 	| $E_{\xi}$ / $F_{\beta}^{w}$ / $F_{\beta}$ / $\mathcal{M}$ 	| $E_{\xi}$ / $F_{\beta}^{w}$ / $F_{\beta}$ / $\mathcal{M}$ 	|
> > | **Ours** 	|                [.786 / .563 / .633 / .093]                	|                [.891 / .725 / .851 / .090]                	|                [.778 / .611 / .705 / .139]                	|
> >
> >
> >
> > --------------
> >
> > **Thanks again for taking the valuable time and providing the insightful comments. Please let us know if you have other questions, and we are happy to address them.**

---

> ### Author Response · Authors · 2021-09-01
> **To Reviewer n5it: we'd like to clear your concerns, and have a further discussion if you have any additional questions.**
>
> Dear Reviewer n5it,
>
> Thanks very much for your efforts in reviewing this paper and providing these valuable questions.
> We sincerely hope that our response has addressed your concerns.
> If there are any remaining concerns, we look forward to discussing them with you here.
>
> Thanks & Regards, \
> Authors of paper-48.

---

### Official Review · Reviewer_DUMJ · 2021-07-16

**Rating:** 6
**Confidence:** 3

**Summary:**

Training saliency detection models with weak supervisions, e.g., image-level tags or captions, is appealing as it removes the costly demand of per-pixel annotations. The manuscript proposed a novel joint semantic mining for weakly supervised RGB-D salient object detection. Specifically, the proposed framework maintain per-pixel pseudo-labels with iterative refinements by reconciling the multimodal input signals. Moreover, the authors also prepared a new dataset that is constructed by augmenting existing benchmark training set with additional image tags and captions. Sufficient experiments on four benchmarks show that the proposed method surpass the state-of-the-art methods. The shortcomings lie in the following aspects: the explanation of the overall framework is not clear enough, take figure 2 as an example, there are too many internal inputs, and the data flow is chaotic.
Please check the following details.

**Limitations And Societal Impact:**

a) The experiments were extensive, but the manuscript placed the ablation experiment at the front of the Experiment section, which was not very friendly for quick understanding of the paper. I suggest the authors analyze the model performance (Sec. 4.4) at the beginning of the Experiments section.

b) The manuscript is well-written, but the explanation of the overall framework is not clear enough, take figure 2 as an example, there are too many internal inputs, and the data flow is chaotic.

c) I wonder how to prevent the model from going down in the wrong direction.

**Main Review:**

a) The main idea is reasonable and novel.
The manuscript proposed a novel joint semantic mining for weakly supervised RGB-D salient object detection. Specifically, the proposed framework maintain per-pixel pseudo-labels with iterative refinements by reconciling the multimodal input signals. Moreover, the authors also prepared a new dataset that is constructed by augmenting existing benchmark training set with additional image tags and captions.
b) Extensive experiments.
In the experiment section, the evaluation on 4 benchmarks demonstrates that the proposed framework attains superior performance to previous models.
c) The overall pipeline is well described and easy to follow.

**Time Spent Reviewing:**

2 hours

---

> ### Author Response · Authors · 2021-08-10
> **Response to Reviewer DUMJ**
>
> **Thanks for your positive comments and valuable suggestions. Your questions will be answered point by point.**
>
> --------------
>
> ***Question1**) The experiments were extensive, but the manuscript placed the ablation experiment at the front of the Experiment section, which was not very friendly for quick understanding of the paper. I suggest the authors analyze the model performance (Sec. 4.4) at the beginning of the Experiments section.*
>
> **Answer1**:
> Thanks for your valuable suggestion. We will reorder the experiment sections accordingly.
>
> --------------
>
> ***Question2**) The manuscript is well-written, but the explanation of the overall framework is not clear enough, take figure 2 as an example, there are too many internal inputs, and the data flow is chaotic.*
>
> **Answer2**:
> We are sorry for the confusion caused in the figures. We will add the meaning of the notations in the caption and improve the presentation, to make the internal inputs and data flow more clear.
>
> --------------
>
> ***Question3**) I wonder how to prevent the model from going down in the wrong direction.*
>
> **Answer3**:
> In this paper, we prevent the model from going down in the wrong direction by iteratively refining the pseudo-labels in training via reconciling the multimodal input signals. First, the SSM is designed to capture the saliency-specific depth semantics, which can be used to effectively eliminate possible background noise and produce the depth-refined pseudo-labels. Second, given these pseudo-labels, our TSM is designed to estimate their confidence scores. The qualitative results of Fig. 7 and the error reduction curves in Fig. 8 demonstrate that the pseudo-label quality is gradually improved as our method is carried out. Moreover, other alternative solutions can be exploited, such as noise-aware losses or curriculum learning.
>
> --------------
>
> **Thanks again for your encouragement and effort, and we will make this paper more clear accordingly. Please let us know if you have other questions.**

---

### Official Review · Reviewer_bzEb · 2021-07-18

**Rating:** 6
**Confidence:** 3

**Summary:**

In this paper authors propose to do saliency object detection with weak supervisions. To train the saliency network, pseudo labels are generated and refined step-by-step, by involving the depth map and the caption completion net. Authors relabeled NJUD and NLPR datasets with off-the-shelf image captioning model. Experiments are carried out on four SOD benchmark datasets and demonstrate the effectiveness of the proposed method.

**Limitations And Societal Impact:**

Authors have adequately addressed the limitations and potential negative societal impact of their work, or there are no obvious issues regarding that direction.

**Main Review:**

This paper proposes to use a weakly-supervised manner to train saliency object detection network, which is an interesting and useful topic. It avoid extra test-time cost so the efficiency of inference is obtained.

The main comparison is made to MSW considering the framework and supervision used in the proposed method. However, this paper and MSW seem to evaluate on different testing datasets and the authors of this paper retrain the MSW model on the new datasets. It would be interesting to have a better aligned evaluation and comparison, if it is possible to reuse some of the datasets used in the previous paper.

The proposed method is also adapted to the fully-supervised settings, and it is interesting to see that the proposed method can obtain higher performance than other state-of-the-arts. More experiments and discussions are necessary to justify why the updated Eq 1 can help with the model training even if we already have the ground-truth supervision.

**Time Spent Reviewing:**

2.5

---

> ### Author Response · Authors · 2021-08-10
> **Response to Reviewer bzEb**
>
> **Thanks for your positive comments and constructive suggestions. Here are our detailed replies to your questions.**
>
> --------------
>
> ***Question1**) The main comparison is made to MSW considering the framework and supervision used in the proposed method. However, this paper and MSW seem to evaluate on different testing datasets and the authors of this paper retrain the MSW model on the new datasets. It would be interesting to have a better aligned evaluation and comparison, if it is possible to reuse some of the datasets used in the previous paper.*
>
> **Answer1**: **Following the suggestion, we evaluated our model on three popular RGB datasets considered in the MSW paper**, with their results reported in the table below.
> Since our model is specifically designed for *RGB-D saliency detection* using cheap weak supervisions, where both the RGB data and depth data are needed for training, we directly use our pretrained model to test on these RGB datasets (benefiting from our additional merit, not relying on depth during inference). For MSW, the saliency maps provided by the authors are used for evaluation. **We will release the source code, pre-trained model and ReadMe.txt**, which allows researchers to be able to easily test our model on other RGB or RGB-D datasets.
>
> | Datasets 	|                         DUT-OMRON                         	|                           ECSSD                           	|                          PASCAL-S                         	|
> |:--------:	|:---------------------------------------------------------:	|:---------------------------------------------------------:	|:---------------------------------------------------------:	|
> |  Metrics 	| $E_{\xi}$ / $F_{\beta}^{w}$ / $F_{\beta}$ / $\mathcal{M}$ 	| $E_{\xi}$ / $F_{\beta}^{w}$ / $F_{\beta}$ / $\mathcal{M}$ 	| $E_{\xi}$ / $F_{\beta}^{w}$ / $F_{\beta}$ / $\mathcal{M}$ 	|
> |    MSW   	|                [.763 / .527 / .609 / .114]                	|                [.884 / .716 / .840 / .096]                	|                [.791 / .619 / .723 / .134]                	|
> | **Ours** 	|                [.786 / .563 / .633 / .093]                	|                [.891 / .725 / .851 / .090]                	|                [.778 / .611 / .705 / .139]                	|
>
>
>
> --------------
>
>
>
> ***Question2**) The proposed method is also adapted to the fully-supervised settings, and it is interesting to see that the proposed method can obtain higher performance than other state-of-the-arts. More experiments and discussions are necessary to justify why the updated Eq 1 can help with the model training even if we already have the ground-truth supervision.*
>
> **Answer2**: Different from existing fully-supervised RGB-D SOD methods that usually design cross-modal fusion strategies (i.e., convolution or skip-connection operations) to integrate RGB and complementary depth features, we innovatively exploit the depth map to construct the spatial supervision signal; then the learned depth semantics is utilized to effectively suppress the background noise in the saliency prediction. **According to your suggestion, in addition to the ablation results in Table 5, we further add more experiments on several existing RGB-D SOD methods by integrating our SSM.** Specifically, we feed the learned depth semantics from the depth network and the saliency prediction from various models (e.g., CTMF, PCA) to the background noise suppression block in SSM to obtain the improved saliency. We report the original results of these methods and the new results of incorporating our SSM (denoted as Ori vs. Our) on the NLPR benchmark in the table below. This further verifies the effectiveness of our approach. In addition to this, we have also visualized the saliency maps of these models, and the quality of saliency map is improved as the SSM is performed. Due to the constraints of rebuttal format, we are unable to add the figure in the rebuttal. **The analyses and the numerical & visual results will be included in the main text.** Thanks for your valuable suggestion.
>
>
>
>
> |     Methods     | [S2MA |  ---] | [CWMN |  ---] | [BBSNet |  ---] | [CPFP |  ---] | [TANet |  ---] | [MMCI |  ---] |  [PCA |  ---] | [CTMF |  ---] |
> |:---------------:|:------:|:----:|:----:|:----:|:----:|:----:|:----:|:----:|:-----:|:----:|:----:|:----:|:----:|:----:|:----:|:----:|
> |        -        |   [Ori  |  **Our**] |  [Ori |   **Our**] |  [Ori |   **Our**] |  [Ori |   **Our**] |  [Ori  |   **Our**] |  [Ori |   **Our**] |  [Ori |   **Our**] |  [Ori |   **Our**] |
> |    $E_{\xi}$    | [.938 | .950] | [.940 | .951] | [.952  | .961] | [.924 | .937] |  [.916 | .938] | [.871 | .910] | [.916 | .929] | [.869 | .918] |
> | $F_{\beta}^{w}$ | [.852 | .873] | [.856 | .879] | [.879  | .897] | [.820 | .841] |  [.789 | .822] | [.688 | .753] | [.772 | .799] | [.691 | .752] |
> |   $F_{\beta}$   | [.853 | .879] | [.859 | .885] | [.882  | .903] | [.822 | .853] |  [.795 | .848] | [.729 | .789] | [.794 | .836] | [.723 | .794] |
> |  $\mathcal{M}$  | [.030 | .025] | [.029 | .023] | [.023  | .021] | [.036 | .029] |  [.041 | .032] | [.059 | .045] | [.044 | .039] | [.056 | .044] |
>
>
>
> --------------
>
> **Thanks again for your appreciation on our work and providing the constructive comments. We hope our response can address your concerns, and please let us know if you have any further questions.**

---

> > ### Comment · Reviewer_bzEb · 2021-09-03
> > **After reading the rebuttal**
> >
> > I have read the response and thank the authors for their effort on addressing my concerns. I believe this paper is of interest to some audiences and decided to keep the original scores.

---

> > > ### Author Response · Authors · 2021-09-03
> > > **Thanks for your positive response**
> > >
> > > Dear Reviewer bzEb,
> > >
> > > We sincerely appreciate your constructive comments and positive response. We will carefully improve the paper and include the above analyses and results in the revised version.
> > >
> > > Thanks & Regards, \
> > > Authors of paper-48.

---

### Author Response · Authors · 2021-08-29
**Paper Discussion**

Dear Reviewers, Area Chairs and Program Chairs,

We are the authors of Paper-48. We sincerely thank the efforts you have made for this paper.

The reviewers were very professional and put forward many insightful questions and valuable suggestions towards improving our paper. In the rebuttal phase, we provided detailed responses to all reviewers' comments point by point, hoping to address the issues raised by reviewers DUMJ, bzEb and n5it.

**The discussion period is coming to an end, and we are now actively awaiting for further discussion from Reviewers**. If you have any questions, we are happy to discuss them with you at any time.

Thanks & Regards, \
Authors of paper-48.

---

### Decision · Program_Chairs · 2021-09-27

**Decision:**

Accept (Poster)

**Comment:**

The paper finally receives mixed reviews with slight preference to being accepted. We thoroughly check the review comments of Reviewer n5it (the only reviewer who has ‘reject’ opinion) and the authors’ responses to them in the rebuttal. Although the n5it submitted no post-rebuttal review, from our perspective, his or her comments mostly fall into technical questions (not some fundamental ones for this work), and the authors’ rebuttal seems to be mostly satisfactory to clarify the concerns. This is the ground for our recommendation, and the authors may need to reflect their own responses about the novelty and additional experiments in the final draft along with the source code.